# HoliGS: Holistic Gaussian Splatting for Embodied View Synthesis

**Xiaoyuan Wang**[1], **Yizhou Zhao**[1], **Botao Ye**[2], **Xiaojun Shan**[3], **Weijie Lyu**[4],
**Lu Qi**[5,*] **Kelvin C.K. Chan**[6], **Yinxiao Li**[6], **Ming-Hsuan Yang**[4,6]

[1]CMU    [2]ETH Zurich    [3]UC San Diego    [4]UC Merced    [5]Insta360    [6]Google DeepMind

## Abstract

We propose HoliGS, a novel deformable Gaussian splatting framework that addresses embodied view synthesis from long monocular RGB videos. Unlike prior 4D Gaussian splatting and dynamic NeRF pipelines, which struggle with training overhead in minute-long captures, our method leverages invertible Gaussian Splatting deformation networks to reconstruct large-scale, dynamic environments accurately. Specifically, we decompose each scene into a static background plus time-varying objects, each represented by learned Gaussian primitives undergoing global rigid transformations, skeleton-driven articulation, and subtle non-rigid deformations via an invertible neural flow. This hierarchical warping strategy enables robust free-viewpoint novel-view rendering from various embodied camera trajectories by attaching Gaussians to a complete canonical foreground shape (*e.g.*, egocentric or third-person follow), which may involve substantial viewpoint changes and interactions between multiple actors. Our experiments demonstrate that HoliGS achieves superior reconstruction quality on challenging datasets while significantly reducing both training and rendering time compared to state-of-the-art monocular deformable NeRFs. These results highlight a practical and scalable solution for EVS in real-world scenarios. The source code will be released.

## 1 Introduction

Understanding and reconstructing dynamic 3D scenes from monocular video remains a fundamental challenge in computer vision, particularly in the context of Embodied View Synthesis (EVS), where camera trajectories dynamically follow actor motions. EVS tasks are crucial for immersive AR/VR experiences, interactive gaming, and robotics, demanding representations capable of handling complex non-rigid deformations, extreme viewpoint changes, and extended temporal sequences.

Despite recent advances in neural rendering for static scenes [1, 2], extending these techniques to dynamic and non-rigid scenarios reveals significant computational and representational challenges. Existing neural radiance fields (NeRF)-based methods [3] face high computational costs during both training and inference, particularly when scaling to minute-long sequences and involving multiple interacting objects. This significantly restricts their practical applicability in real-time environments.

Gaussian Splatting (GS) approaches [2], known for efficient rendering in static scenes through compact anisotropic Gaussian primitives, also encounter limitations in dynamic contexts. Current deformable Gaussian Splatting techniques [4, 5] are typically constrained to short-duration captures or scenarios with minimal non-rigid motion. When applied to EVS tasks involving intricate interactions, these methods yield inconsistent reconstructions with noticeable artifacts(see Figure 2).

---

*Corresponding author.

39th Conference on Neural Information Processing Systems (NeurIPS 2025).

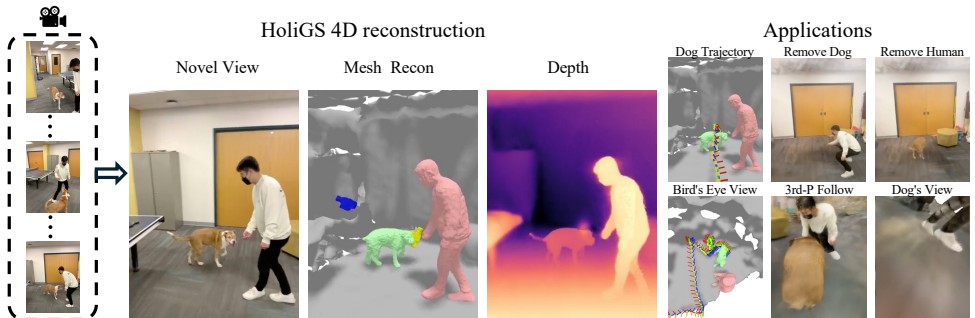

Figure 1: **Overview.** From a phone capture of humans and animals in motion, HoliGS reconstructs temporally consistent geometry, appearance, and depth, enabling novel-view synthesis, deformable mesh recovery, and dense depth estimation. These reconstructions support a range of embodied applications, including actor-specific view synthesis (*e.g.*, third-person and egocentric perspectives), object-specific removal, and actor-centric visualization (*e.g.*, dog's-eye view). HoliGS also enables spatiotemporal behavior analysis such as trajectory visualization.

| Method | Entire Scenes | Deform. Objects | Global 6-DOF Traj. | Long Videos | Extreme Views | Fast Rendering |
|---|---|---|---|---|---|---|
| BANMo | ✗ | ✓ | ✗ | ✓ | ✓ | ✗ |
| RAC | ✗ | ✓ | ✗ | ✓ | ✓ | ✗ |
| Vidu4D | ✗ | ✓ | ✗ | ✓ | ✓ | ✓ |
| MoSca | ✓ | ✓ | ✗ | ✗ | ✗ | ✓ |
| SoM | ✓ | ✓ | ✗ | ✗ | ✗ | ✓ |
| SC-GS | ✓ | ✓ | ✗ | ✗ | ✗ | ✓ |
| Dyn.Guss | ✓ | ✓ | ✗ | ✓ | ✗ | ✓ |
| G.Marbles | ✓ | ✓ | ✗ | ✓ | ✗ | ✓ |
| Total-Recon | ✓ | ✓ | ✓ | ✓ | ✓ | ✗ |
| Ours | ✓ | ✓ | ✓ | ✓ | ✓ | ✓ |

Table 1: **Comparison to Related Work.** HoliGS targets embodied view synthesis of dynamic scenes and process *minute-long videos*, and render *extreme views*.

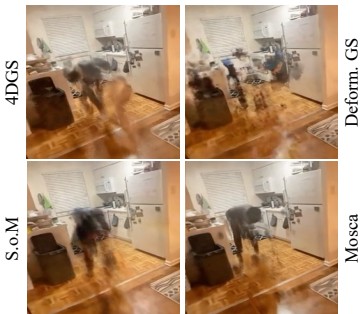

Figure 2: Performance of SOTA methods.

Furthermore, several existing methods [6, 7, 8] rely heavily on off-the-shelf point-tracking models [5], introducing significant computational overhead and exhibiting fragility under severe occlusions. These methods also fail to generalize effectively to arbitrary viewpoint trajectories essential for comprehensive EVS scenarios, severely limiting their utility in real-world conditions marked by frequent occlusions and the need for viewpoint flexibility.

To overcome these critical limitations, we propose HoliGS, a holistic Gaussian Splatting method explicitly designed for EVS applications. Unlike previous methods, our framework introduces a Gaussian-based deformation model that directly manages articulated non-rigid transformations without relying on traditional tracking pipelines. This innovation ensures consistent and artifact-free reconstructions across complex sequences involving human and animal interactions.

Specifically, our approach includes a novel deformable Gaussian Splatting pipeline and an optimized strategy to maintain high-quality rendering under extreme viewpoint variations, such as egocentric, third-person follow, and overhead perspectives. Additionally, we integrate an invertible deformation model, enabling stable reconstructions over prolonged durations without sacrificing efficiency.

Extensive experimental evaluation demonstrates that HoliGS significantly outperforms state-of-the-art methods in terms of both rendering quality and computational speed, achieving real-time rendering capabilities on consumer hardware. Our results confirm robust performance across diverse, challenging, dynamic sequences featuring multiple interacting entities and complex articulated motions, scenarios where prior techniques either fail or produce substantial visual artifacts. The main contributions of this work are:

- We introduce a holistic Gaussian Splatting method for EVS tailored to 6-DOF embodied camera paths, outperforming existing state-of-the-art approaches [3, 9].
- We propose an invertible deformation model that ensures stable reconstruction over extended periods without compromising computational efficiency.
- We evaluate our model on diverse challenging dynamic scenes against existing methods and show that our approach achieves robust view synthesis and scalable to minute-long videos.

## 2  Related Work

**Dynamic Scene Reconstruction.** Reconstructing dynamic scenes from videos has been an active research area, traditionally relying on multi-view stereo systems [10, 11, 12, 13, 14, 15, 16, 17, 18, 19, 20, 21, 22, 23]. Recently, another series of works focus on monocular scene reconstruction methods [24, 25, 26, 27, 28, 29, 30, 31, 32, 33, 34, 35, 36, 37, 38, 39, 40, 41, 42, 43]. Dynamic methods often utilize either temporal conditioning as an additional input dimension [44] or canonical-space representations with deformation fields [45, 25, 46]. Grid-based representations [47, 48] have further accelerated these methods, enabling efficient optimization for dynamic scene reconstruction [17, 49, 50]. Despite significant progress, these approaches still suffer from high computational costs, especially in real-time and long video scenarios with complex motion patterns or prolonged video sequences.

**Embodied View Synthesis (EVS).** EVS introduces additional complexity, requiring representations capable of handling camera trajectories that closely follow or interact with dynamic subjects. Existing methods like DyCheck [51] highlight the inadequacies of current benchmarks, which often do not accurately reflect realistic everyday scenarios involving limited viewpoints and complex dynamics. Methods designed specifically for monocular EVS [52, 3, 53] aim to mitigate these issues through hybrid representations or generative methods. Nevertheless, these methods typically rely heavily on domain-specific priors or computationally intensive tracking modules, restricting their robustness under occlusions and generalization across diverse view trajectories.

**Articulated Object Reconstruction.** Articulated object reconstruction, especially for humans and animals, often utilze parametric templates [54, 55, 56, 57, 58], which impose strong geometric priors and facilitate reconstruction from sparse views or monocular videos [59, 60, 61, 62]. However, these models typically struggle with capturing personalized or detailed appearance variations. More recent non-parametric neural methods have combined neural radiance fields with articulated models [63, 64, 65, 66, 67, 68, 69, 70, 71], capturing richer detail but at a significant computational cost. Our method diverges by directly modeling articulated motion without relying on predefined parametric templates, instead employing a flexible Gaussian-based deformation model optimized for dynamic reconstruction.

**Non-Rigid Structure from Motion.** Non-rigid Structure from Motion (NRSfM) aims to reconstruct the 3D shape and deformation of objects from monocular videos, handling scenarios where scene points undergo complex, articulated, or continuous deformation. Traditional SfM and visual SLAM methods [72, 73, 74] typically assume static environments, enforcing strict epipolar constraints unsuitable for dynamic scenes. Recent methods address this limitation by jointly estimating camera poses, scene geometry, and deformation fields [75, 76]. These approaches, however, often rely on time-intensive test-time optimization or explicit motion segmentation, limiting their scalability and efficiency. Differently, our method leverages a Gaussian-based deformation model to explicitly encode articulated non-rigid transformations, enabling efficient reconstruction without the need for computationally costly per-video fine-tuning or explicit motion segmentation. This approach facilitates robust reconstruction of dynamic interactions in everyday monocular videos, effectively overcoming challenges posed by occlusions and extensive deformation.

The proposed framework, HoliGS, combines the advantages of articulated object reconstruction and static Gaussian Splatting to enable efficient, high-quality embodied view synthesis for dynamic scenes captured from monocular videos, overcoming limitations associated with existing methods.

## 3  Method

In this section, we introduce HoliGS, a hierarchical 4D representation that models dynamic scenes as the union of a static background and time-varying deformable objects. Our framework leverages Gaussian Splatting to represent both the static and dynamic components and employs a series of invertible warping operations to capture articulated and non-rigid deformations. The final scene at time $t$ is given by $\mathcal{S}(t)=\mathcal{G}(t) \cup \mathcal{H}$, where $\mathcal{H}$ is the set of static background Gaussians and $\mathcal{G}(t)$ contains the dynamic, time-varying Gaussians splitting articulated foreground objects.

### 3.1  Hierarchical Dynamic Warping

To robustly model motion ranging from whole-body translations to fabric flutter, we use a two-stage warping strategy. At a glance, large articulated displacements are first explained by a skeleton-driven

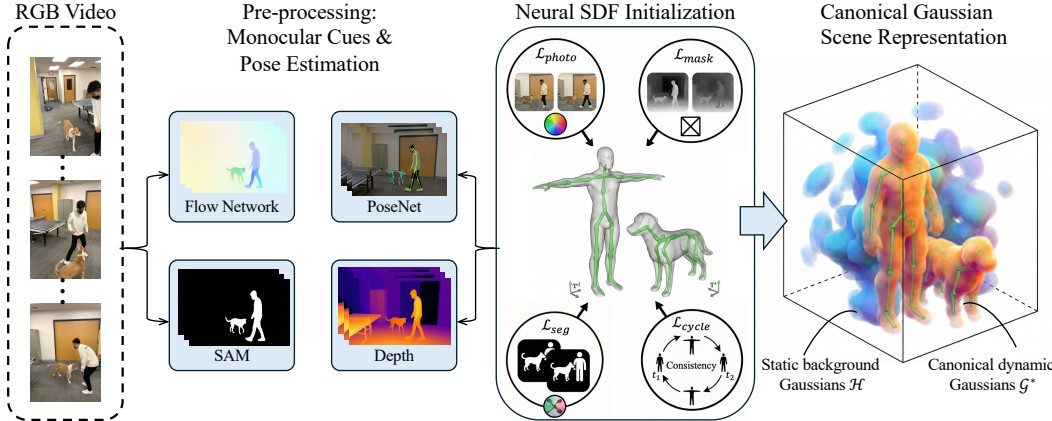

Figure 3: **HoliGS Pipeline.** *Warping network initialization*: We jointly optimize poses, articulation, soft deformation, and in a neural SDF proxy to obtain a fast converging deformation field that provides a strong starting point for Gaussian splitting. *Deformable GS*: the objective is switched to dynamic Gaussian splatting, and the deformed foreground is composited with the static background to yield the final 4D scene.

transform, after which a soft, flow-based deformation field refines any residual non-rigid detail. All derivations and exact matrix expressions are deferred to the supplementary material.

**Global movements.** Every video frame is aligned to the camera via two rigid $SE(3)$ transforms: the *background-to-camera* map $G_b$ and the *object-root-to-camera* map $G_o$. Both transforms are regressed by lightweight Fourier MLPs that output six twist parameters per frame, giving us frame-specific poses without needing an external tracker.

**Skeleton-driven warping.** The core articulated motion is handled by a bone hierarchy with $B$ bones. Each bone $b$ has a static reference pose $(c_b^*, V_b^*, \Lambda_b^*)$ encoding center, rotation, and scale, respectively. At time $t$, a learned twist vector $\hat{\eta}_b(t) \in SE(3)$ is exponentiated to produce the bone pose $J_b(t)$. We measure how much a 3-D point $P_k^*$ belongs to each bone by a Mahalanobis distance in the bone's scaled–rotated frame; a softmax over these distances yields skinning weights $w(t)$. Dual-quaternion blend skinning (DQB) [77] fuses the individual bone transforms into a single $SE(3)$ map $J(t)$, which is then applied to every Gaussian center, rotation, and scale. Conceptually, this step captures all "rigid-but-articulated" effects such as limbs, torsos, or tails.

**Soft deformation field.** After skeletal warping, many objects still exhibit subtle surface changes—loose clothing, hair swaying, muscle bulges—that cannot be explained by rigid bones. We address this with a *soft deformation field* $S(\cdot, \omega_d)$ implemented as an invertible RealNVP flow [78]. Given a canonical point $X$ and a per-frame latent code $\omega_d$, the field outputs a refined position $X' = S(X, \omega_d)$. Invertibility guarantees that $S^{-1}$ exists; we therefore impose a 3-D cycle-consistency loss: $\mathcal{L}_{\mathrm{cyc}} = \|S^{-1}(S(X, \omega_d), \omega_d) - X\|_2^2$, which forces the forward and reverse mappings to cancel out and stabilizes training. Because the flow operates in a *fixed* canonical space, it never has to chase a moving target, allowing it to converge quickly even when the deformations are highly nonlinear.

**Why hierarchy matters.** Articulated bones give the model an inductive bias toward plausible large-scale motion, while the soft field soaks up the remaining fine detail. Each module solves a simpler task and therefore converges faster than a single, monolithic deformation network. Empirically, the skeletal stage explains $\approx 90\%$ of visible motion energy, leaving only low-amplitude corrections to the RealNVP field. Full mathematical details—the Lie-algebra twist representation, the exact Mahalanobis weighting, and the DQB formulation—are provided in the supplementary materials.

**Combined warping pipeline.** Integrating the above components, a point $X^*$ in canonical space is warped to its dynamic position at time $t$ according to:

$$ X^t \;=\; G_o^{t\,-1} \cdot J^{t\,-1} \cdot S^{-1}\Big(X^*, \omega_d^t\Big). \tag{1} $$

Inspired by Omnimotion [79], HoliGS also enables a forward warp

$$ X^* \;=\; S \cdot J^t \cdot G_o^t\Big(X^t, \omega_d^t\Big). \tag{2} $$

This unified warping function seamlessly integrates global, skeletal articulation, and fine-scale deformations, enabling our framework to render high-quality 4D scenes with complex dynamics.

## 3.2 Deformation Network Initialization

For our dynamic scene representation, we establish initial transformation parameters by pre-training a neural SDF that warps sampled points on camera rays from the static state to the warped states, similar to [68]. We apply Posenet [80] to obtain the rigid-body transformations $T^d$ and time-dependent skeletons for each deformable object in the scene. This network provides robust pose estimates even under challenging viewing conditions. Concurrently, we initialize the background component transformations $T^s$ using camera pose information extracted from the capture device's motion sensors. This hybrid initialization strategy ensures stable convergence during subsequent optimization stages while accommodating both foreground dynamic objects and static background elements within our unified representation. Then, we initialize the foreground Gaussian point cloud from the pre-trained neural SDF by sampling points on its surface, with objective function:

$$\mathcal{L} = \underbrace{\mathcal{L}_{\text{photo}}}_{\substack{\text{photometric}\\\text{consistency}}} + \underbrace{\lambda_{\text{depth}}\mathcal{L}_{\text{depth}} + \lambda_{\text{SDF}}\mathcal{L}_{\text{SDF}}}_{\substack{\text{geometric}\\\text{constraints}}} + \underbrace{\lambda_{\text{flow}}\mathcal{L}_{\text{flow}} + \lambda_{\text{cycle}}\mathcal{L}_{\text{cycle}}}_{\substack{\text{motion}\\\text{consistency}}} + \underbrace{\mathcal{L}_{\text{seg}}}_{\substack{\text{mask}\\\text{supervision}}} . \tag{3}$$

Here, the photometric loss $\mathcal{L}_{\text{photo}}$ enforces appearance consistency. For geometry constraints: the depth term $\mathcal{L}_{\text{depth}} = \sum_{p^t} \|D(p^t) - \hat{D}(p^t)\|_2^2$ aligns our predicted depth $\hat{D}$ with an off-the-shelf monocular depth estimator $D$ [81], promoting correct scene scale, and the SDF term $\mathcal{L}_{\text{SDF}} = \sum_{X_i^t}(\|\nabla_{X_i^t}\Phi_{\text{SDF}}(X_i^t)\|_2 - 1)^2$ enforces the signed distance field $\Phi_{\text{SDF}}$ to behave like a true distance function by constraining its gradient norm to one. Motion consistency is imposed by flow loss $\mathcal{L}_{\text{flow}} = \sum_{p^t} \|V(p^t) - \hat{V}(p^t)\|_2^2$ and cycle loss where

$$\mathcal{L}_{\text{cycle}} = \sum_{i,j} \lambda_j \, \beta_{i,j} \, \|\mathcal{F}_{\text{fwd},j}^{t'}(\mathcal{F}_{\text{bwd},j}^t(X_i^t)) - X_i^t\|_2^2 \tag{4}$$

weighted by importance factors $\lambda_j$ and $\beta_{i,j}$, aligning RAFT optical flow [82] and satisfying forward–backward cycle consistency. Finally, segmentation supervision is given by $\mathcal{L}_{\text{seg}} = \sum_{p^t} \|M_{\text{pred}}(p^t) - M_{\text{gt}}(p^t)\|_2^2$, with $M_{\text{gt}}$ obtained from SAM [83]. $p^t \in \mathbb{R}^2$ represents pixel coordinates at time $t$, $X_i^t \in \mathbb{R}^3$ denotes the $i$-th sample point in world space corresponding to $X_i^t \in \mathbb{R}^3$ in camera space. Weights $\{\lambda_{\text{depth}}, \lambda_{\text{SDF}}, \lambda_{\text{flow}}, \lambda_{\text{cycle}}\}$ are tuned to balance these complementary constraints.

## 3.3 Deformable Gaussian Splatting Optimization Objectives

Our composite Gaussian Splatting representation incorporates $N$ scene elements, global transformation matrices $T_t^i$, and bidirectional deformation fields $F_{\text{forward}}^i$ and $F_{\text{backward}}^i$. The optimization process integrates multiple objectives to ensure high-quality reconstruction and temporal consistency:

$$\mathcal{L} = \mathcal{L}_{\text{photo}} + \mathcal{L}_{\text{depth}} + \mathcal{L}_{\text{seg}} + \mathcal{L}_{\text{normal}}. \tag{5}$$

Besides the loss terms we explained in initialization, $\mathcal{L}_{\text{photo}}$, $\mathcal{L}_{\text{depth}}$, and $\mathcal{L}_{\text{seg}}$, we incorporate additional normal supervision to align the estimated entire scene surface normals with observed ones $\mathcal{L}_{\text{normal}} = \sum_{p^t} \|N(p^t) - \hat{N}(p^t)\|^2$. This comprehensive optimization framework ensures geometric accuracy, appearance fidelity, and temporal consistency in our dynamic scene representation.

## 3.4 Embodied View Synthesis

To effectively perform EVS, HoliGS transforms dynamic 3D Gaussian primitives into consistent, egocentric viewpoints that naturally follow the motion of articulated objects, such as humans and animals. Specifically, for each Gaussian primitive, we apply a forward warping function $W_{t\to j}$ : $X^* \to X_t$, which maps points from a canonical space $X^*$ to the deformed configuration at time $t$. This deformation accounts explicitly for non-rigid articulated transformations, ensuring accurate representation of complex motions such as limb articulations or interactions among multiple entities.

Subsequently, to achieve embodied viewpoints, we employ a rigid-body transformation $G_t^0$, positioning the virtual egocentric camera within the world coordinate system. It aligns the viewer's

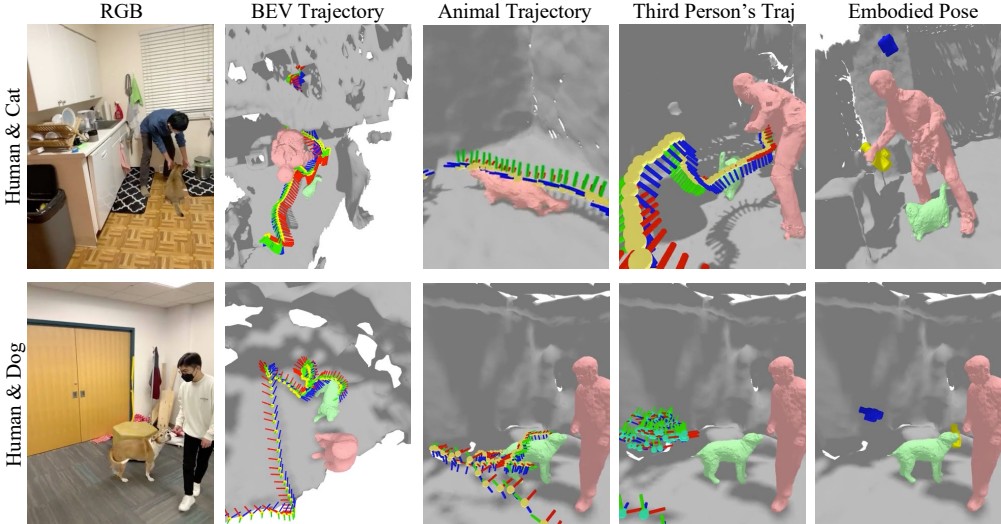

Figure 4: **Foreground Embodied Trajectory.** For two challenging sequences, HumanCat and HumanDog, we show: (i) the joint bird's-eye-view (BEV) trajectory of a foreground actor, (ii) the articulated animal trajectory, (iii) the articulated human trajectory, and (iv) both objects' embodied camera pose. Our method recovers smooth, collision-free paths that faithfully follow each actor while remaining mutually consistent, enabling stable first-person or over-the-shoulder replays for complex multi-agent interactions.

perspective with the foreground, enabling realistic rendering of scenarios such as first-person views or third-person perspectives following actors in motion (illustrated in Figure 4).

By integrating the deformation network, our method reliably synthesizes novel embodied viewpoints that remain coherent across complex motions. Our unified Gaussian-based deformation and viewpoint adjustment strategy significantly simplifies optimization and achieves near real-time performance. This enables practical usage in interactive AR/VR applications, immersive gaming experiences, and robotics, where rapid viewpoint changes and accurate motion tracking are essential.

## 4 Experiments

### 4.1 Training and Optimization

We adopt a two-phase procedure to optimize our dynamic Gaussian representation: *Component pre-training* and *joint refinement*. During pre-training, each component (e.g., a deformable object or the static background) is optimized separately. Once pre-training is completed, all components are combined for joint refinement using color, depth, normal, and mask objectives. Training follows standard Gaussian Splatting protocols [2]. The synergy between our deformation-centric design and the parametric Gaussian framework accelerates convergence considerably. On NVIDIA H20 GPUs, each pre-training or refinement stage completes in about 30 minutes, enabling full scenes (including multiple deformable objects) to converge in two hours, significantly faster than other approaches.

**Component pre-training.** We initialize the deformation network by minimizing the overall loss (3), with default weights set as: $\lambda_{\text{depth}} = 5$ (or 1.5 for the HUMAN 1 sequence), $\lambda_{\text{color}} = 0.1$, $\lambda_{\text{flow}} = 1$, $\lambda_{\text{cycle}} = 1$, and $\lambda_{\text{segment}} = 1$. This eikonal term is weighted by $\lambda_{\text{SDF}} = 0.001$ to ensure proper geometric properties. For this computation, we sample 17 uniformly distributed points $X_i^t$ along each camera ray $r^t$ centered at the surface point derived from back-projecting the ground-truth depth.

**Joint fine-tuning.** During the joint optimization phase, we simultaneously refine all object representations by minimizing loss (5) for an additional 6,000 iterations. The default weights for these objectives are $\lambda_{\text{photo}} = 1$, $\lambda_{\text{normal}} = 1$, $\lambda_{\text{depth}} = 5$, and $\lambda_{\text{seg},j} = 1$. By default, we freeze the background's appearance and geometry parameters while allowing optimization of its global transformation $T_0^b$, the foreground objects' transformations $T_t^f$, and the foreground appearance and geometry parameters (for HUMAN 1, we use $\lambda_{\text{depth}} = 1.5$), we allow background appearance and geometry optimization during joint fine-tuning). This joint fine-tuning phase significantly enhances the visual coherence of foreground elements and improves the modeling of inter-object interactions.

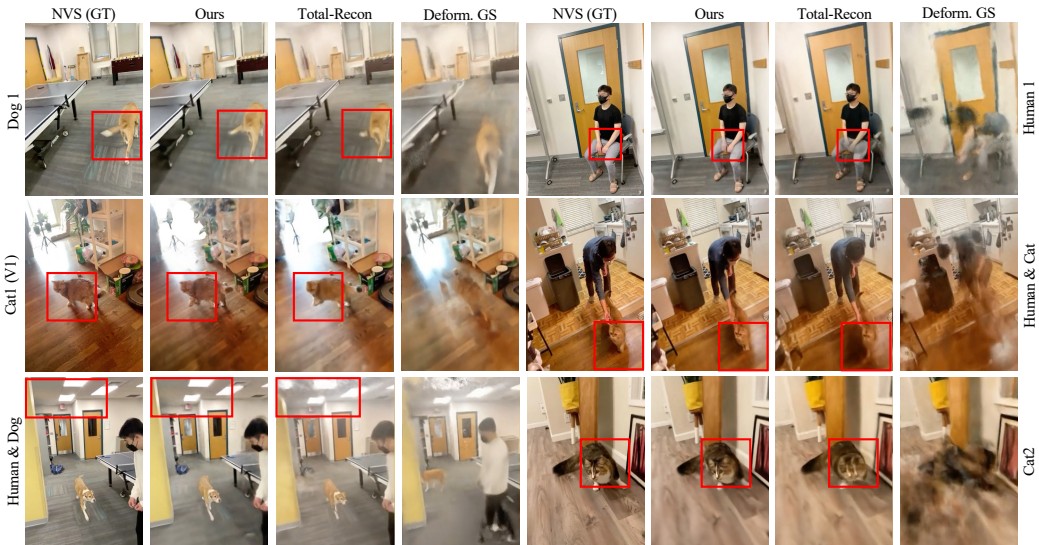

Figure 5: **Baseline Comparison.** We qualitatively compare HoliGS against four SOTA baselines and a direct NVS ground-truth reference across *Dog 1*, *Cat 1*, *Human 1*, and the challenging multi-actor *Human 2 & Cat* sequences. Each column shows photometric renderings (top) and corresponding depth reconstructions (bottom). Red inset boxes highlight the most error-prone regions for articulated motion and occlusion (e.g. tail swing, paw lift, garment folds, and human–animal interaction). Compared with baselines, HoliGS better preserves fine-grained appearance and yields geometrically consistent depth maps with fewer tearing or bleeding artifacts—especially under large viewpoint changes and prolonged, highly non-rigid deformations.

| | Dog 1 (v1) (626 images) | | | Dog 1 (v2) (531 images) | | | Cat 1 (v1) (641 images) | | | Cat 1 (v2) (632 images) | | | Cat 2 (v1) (834 images) | | | Cat 2 (v2) (901 images) | | |
|---|---|---|---|---|---|---|---|---|---|---|---|---|---|---|---|---|---|---|
| | LPIPS↓ | PSNR↑ | SSIM↑ | LPIPS↓ | PSNR↑ | SSIM↑ | LPIPS↓ | PSNR↑ | SSIM↑ | LPIPS↓ | PSNR↑ | SSIM↑ | LPIPS↓ | PSNR↑ | SSIM↑ | LPIPS↓ | PSNR↑ | SSIM↑ |
| HyperNeRF | .634 | 12.84 | .673 | .432 | 14.27 | .721 | .521 | 14.86 | .632 | .438 | 14.87 | .597 | .641 | 12.32 | .632 | .397 | 15.68 | .657 |
| D²NeRF | .540 | 13.37 | .694 | .546 | 11.74 | .685 | .687 | 10.92 | .545 | .588 | 11.88 | .548 | .556 | 12.55 | .664 | .595 | 12.71 | .604 |
| HyperNeRF (w/ depth) | .373 | 16.86 | .730 | .425 | 16.95 | .740 | .532 | 14.37 | .621 | .371 | 15.65 | .617 | .330 | 18.47 | .728 | .376 | 16.56 | .670 |
| D²NeRF (w/ depth) | .507 | 13.44 | .698 | .532 | 11.88 | .690 | .685 | 10.81 | .534 | .580 | 12.00 | .563 | .561 | 12.59 | .656 | .553 | 12.76 | .629 |
| Total-Recon (w/ depth) | .271 | 17.60 | .745 | .313 | 17.78 | .768 | .382 | 15.77 | .657 | .333 | 16.44 | .652 | .237 | 21.22 | .793 | .281 | 18.52 | .713 |
| Deformable-gs (w/ depth) | .520 | 12.35 | .432 | .490 | 12.78 | .450 | .565 | 11.92 | .398 | .530 | 12.30 | .410 | .600 | 11.50 | .380 | .510 | 12.60 | .420 |
| 4DGS (w/ depth) | .525 | 12.40 | .425 | .495 | 12.65 | .445 | .570 | 11.85 | .390 | .535 | 12.25 | .415 | .605 | 11.45 | .375 | .515 | 12.55 | .430 |
| GS-marble | —— | OOM | —— | .530 | 12.45 | .430 | —— | OOM | —— | —— | OOM | —— | —— | OOM | —— | —— | OOM | —— |
| MoSca | —— | OOM | —— | .312 | 19.95 | .695 | —— | OOM | —— | —— | OOM | —— | —— | OOM | —— | —— | OOM | —— |
| Shape-of-Motion | —— | OOM | —— | .282 | 20.85 | .785 | —— | OOM | —— | —— | OOM | —— | —— | OOM | —— | —— | OOM | —— |
| **Ours** | **.251** | **20.12** | **.825** | **.285** | **21.37** | **.791** | **.319** | **20.52** | **.711** | **.285** | **21.74** | **.693** | **.203** | **22.94** | **.693** | **.262** | **22.07** | **.763** |

| | Cat 3 (767 images) | | | Human 1 (550 images) | | | Human 2 (483 images) | | | Human - Dog (392 images) | | | Human - Cat (431 images) | | | Mean | | |
|---|---|---|---|---|---|---|---|---|---|---|---|---|---|---|---|---|---|---|
| | LPIPS↓ | PSNR↑ | SSIM↑ | LPIPS↓ | PSNR↑ | SSIM↑ | LPIPS↓ | PSNR↑ | SSIM↑ | LPIPS↓ | PSNR↑ | SSIM↑ | LPIPS↓ | PSNR↑ | SSIM↑ | LPIPS↓ | PSNR↑ | SSIM↑ |
| HyperNeRF | .592 | 13.74 | .624 | .632 | 11.94 | .603 | .585 | 14.97 | .620 | .487 | 15.04 | .699 | .462 | 13.52 | .512 | .531 | 14.00 | .635 |
| D²NeRF | .759 | 11.03 | .578 | .588 | 11.88 | .638 | .630 | 12.13 | .599 | .576 | 12.41 | .652 | .628 | 10.41 | .453 | .611 | 11.97 | .608 |
| HyperNeRF (w/ depth) | .514 | 14.86 | .635 | .501 | 13.25 | .664 | .445 | 15.58 | .665 | .450 | 15.01 | .704 | .456 | 14.40 | .535 | .428 | 15.80 | .667 |
| D²NeRF (w/ depth) | .730 | 11.08 | .582 | .585 | 12.14 | .638 | .609 | 12.11 | .612 | .608 | 12.30 | .633 | .645 | 10.51 | .451 | .599 | 12.02 | .611 |
| Total-Recon (w/ depth) | .261 | 19.89 | .734 | .213 | 18.39 | .778 | .264 | 16.73 | .712 | .256 | 16.69 | .756 | .233 | 17.67 | .630 | .278 | 18.11 | .724 |
| Deformable-gs (w/ depth) | .550 | 12.45 | .410 | .505 | 12.80 | .430 | .560 | 11.95 | .400 | .540 | 12.10 | .420 | .590 | 11.70 | .390 | .542 | 12.22 | .413 |
| 4DGS (w/ depth) | .545 | 12.50 | .415 | .510 | 12.75 | .435 | .565 | 11.90 | .405 | .535 | 12.15 | .425 | .595 | 11.65 | .385 | .545 | 12.19 | .413 |
| GS-marble | —— | OOM | —— | .548 | 12.50 | .415 | .545 | 12.08 | .405 | .538 | 12.32 | .418 | .580 | 11.85 | .399 | —— | NA | —— |
| MoSca | —— | OOM | —— | —— | OOM | —— | .263 | 18.15 | .711 | **.241** | 21.10 | **.781** | .243 | 19.05 | .730 | —— | NA | —— |
| Shape-of-Motion | —— | OOM | —— | .214 | 18.45 | .776 | .262 | 16.78 | .715 | .253 | 16.75 | .758 | .235 | 17.55 | .635 | —— | NA | —— |
| **Ours** | **.247** | **20.50** | **.744** | **.211** | **20.19** | **.782** | **.251** | **18.78** | **.725** | **.247** | 20.56 | .776 | **.229** | **21.34** | **.688** | **.263** | **21.31** | **.747** |

Table 2: **Quantitative Comparisons on Novel View Synthesis (Visual Metrics)**. We compare our method to previous dynamic NVS works and their depth-supervised variants on the 11 sequences of our stereo RGB dataset in terms of LPIPS, PSNR, and SSIM. Our method significantly outperforms all baselines for all sequences.

## 4.2 Qualitative and Quantitative Results

Figure 5 shows representative visualizations comparing the photometric and depth reconstruction quality of HoliGS against Total-Recon [3], Deformable GS [84], and 4DGS [85]. These results demonstrate the superior performance of our method under various challenging conditions.

Quantitative results for novel view synthesis are reported in Tables 2 and 3. Table 2 presents visual metrics across the Total-Recon dataset, while Table 3 reports depth accuracy metrics (Acc@0.1m and RMS depth error). Our method consistently outperforms the baselines in both sets of metrics.

| | DOG 1 | | DOG 1 (v2) | | CAT 1 | | CAT 1 (v2) | | CAT 2 | | CAT 2 (v2) | | CAT 3 | | HUMAN 1 | | HUMAN 2 | | HUMAN - DOG | | HUMAN - CAT | | MEAN | |
|---|---|---|---|---|---|---|---|---|---|---|---|---|---|---|---|---|---|---|---|---|---|---|---|---|
| | Acc↑ | $\epsilon_{depth}$↓ | Acc↑ | $\epsilon_{depth}$↓ | Acc↑ | $\epsilon_{depth}$↓ | Acc↑ | $\epsilon_{depth}$↓ | Acc↑ | $\epsilon_{depth}$↓ | Acc↑ | $\epsilon_{depth}$↓ | Acc↑ | $\epsilon_{depth}$↓ | Acc↑ | $\epsilon_{depth}$↓ | Acc↑ | $\epsilon_{depth}$↓ | Acc↑ | $\epsilon_{depth}$↓ | Acc↑ | $\epsilon_{depth}$↓ | Acc↑ | $\epsilon_{depth}$↓ |
| HyperNeRF | .107 | .687 | .176 | .870 | .316 | .476 | .314 | .564 | .277 | .765 | .252 | .811 | .213 | .800 | .053 | .821 | .067 | 1.665 | .072 | .894 | .162 | .862 | .198 | .855 |
| D²NeRF | .219 | .463 | .220 | .456 | .346 | .334 | .403 | .314 | .333 | .371 | .339 | .361 | .231 | .523 | .066 | 1.063 | .128 | .890 | .078 | .847 | .126 | .880 | .247 | .739 |
| HyperNeRF | .352 | .331 | .357 | .338 | .552 | .206 | .596 | .209 | .605 | .154 | .612 | .170 | .451 | .285 | .211 | .591 | .249 | .611 | .283 | .565 | .214 | .613 | .439 | .374 |
| D²NeRF | .338 | .423 | .270 | .445 | .510 | .325 | .362 | .313 | .438 | .298 | .376 | .318 | .243 | .496 | .086 | .984 | .131 | .813 | .154 | .789 | .176 | .757 | .302 | .549 |
| Total-Recon | .841 | .165 | .790 | .167 | **.889** | **.184** | .894 | .124 | .967 | .050 | .925 | .081 | .949 | .066 | .909 | .142 | .849 | .142 | .827 | .204 | .914 | .104 | .895 | .131 |
| Def.GS | .172 | .599 | .183 | .612 | .320 | .415 | .328 | .432 | .295 | .485 | .271 | .494 | .225 | .598 | .070 | .912 | .109 | .940 | .085 | .862 | .145 | .795 | .215 | .632 |
| 4DGS | .175 | .603 | .178 | .620 | .315 | .423 | .325 | .436 | .292 | .481 | .268 | .499 | .232 | .592 | .073 | .908 | .113 | .936 | .089 | .859 | .142 | .802 | .200 | .651 |
| GS-marble | —— | OOM | .180 | .615 | —— | OOM | —— | OOM | —— | OOM | —— | OOM | —— | OOM | .175 | .710 | .210 | .838 | .187 | .801 | .143 | .799 | —— | NA |
| MoSca | —— | OOM | .792 | .165 | —— | OOM | —— | OOM | —— | OOM | —— | OOM | —— | OOM | .850 | .141 | .826 | .205 | | | .912 | .106 | —— | NA |
| S.o.M | —— | OOM | .788 | .168 | —— | OOM | —— | OOM | —— | OOM | —— | OOM | —— | OOM | .908 | .144 | .845 | .145 | .825 | .206 | .911 | .108 | —— | NA |
| **Ours** | **.845** | **.160** | **.795** | **.163** | .880 | .190 | **.898** | **.122** | **.970** | **.048** | **.928** | **.079** | **.955** | **.064** | **.915** | **.138** | **.855** | **.139** | **.830** | .202 | **.920** | **.102** | **.901** | **.127** |

Table 3: **Quantitative Comparisons on Novel View Synthesis (Depth Metrics)**. We compare HoliGS to previous works on the Total-Recon dataset in terms of the average accuracy at 0.1m (Acc@0.1m) and the RMS depth error $\epsilon_{depth}$ (units: meters). Our method significantly outperforms all baselines for all sequences.

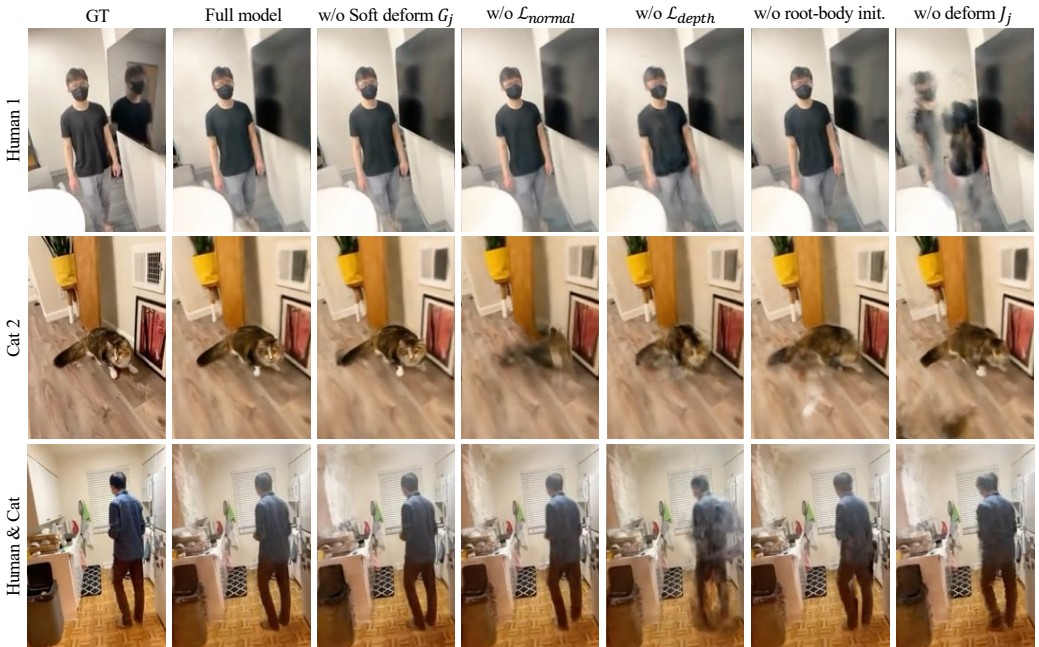

Figure 6: **Ablation Studies Visualization.** Qualitative impact of removing soft deformation, normal/depth supervision, root–body initialization, and skeleton deformation. Each omission introduces increasing blur, drift, and silhouette break-up, whereas the full model remains sharp and stable.

Table 4 and Figure 6 evaluate the contribution of each deform component systematically removing key elements: the depth supervision, the normal supervision, the deformation field $F^t$, soft deformation $S$, pose initialization from external estimators, and the rigid transformation $T_j^t$, where $j$ identifies a deformable object. For all ablations, we maintain the same core optimization objectives used in our full method while initializing camera parameters $T_b^t$ from device sensors. For configurations without rigid body modeling, we initialize each object's pose with predictions from PoseNet and optimize them during reconstruction; for row 6, we replace these predictions with identity transformations.

**Geometric supervision.** Table 4 demonstrates that removing depth supervision (row 2) significantly reduces average accuracy. Figure 6 reveals that this stems from scale inconsistency between objects - while removing depth supervision does not severely impact training-view RGB renderings, it introduces critical failure modes in novel-view reconstructions: (a) floating foreground objects, evidenced by misaligned shadows, and (b) incorrect occlusion relationships between subjects. Without depth supervision, our method overfits to training perspectives and produces a degenerate scene representation where objects fail to maintain consistent scale relationships.

### 4.3 Ablations Studies

Similarly, our results show that normal supervision (row 3) provides crucial geometric guidance. Without normal constraints, the model struggles to capture fine surface details and produces less

| Methods | Depth Loss | Normal Loss | Deform. Obj. | Root Init. | Root Motion | Deform. Soft | LPIPS↓ | Acc@0.1m↑ |
|---|---|---|---|---|---|---|---|---|
| (1) **Full model** | ✓ | ✓ | ✓ | ✓ | ✓ | ✓ | **.263** | **.896** |
| (2) w/o loss $\mathcal{L}_{depth}$ | ✗ | ✓ | ✓ | ✓ | ✓ | ✓ | .385 | .847 |
| (3) w/o loss $\mathcal{L}_{normal}$ | ✓ | ✗ | ✓ | ✓ | ✓ | ✓ | .288 | .832 |
| (4) w/o deform. $\mathbf{J}_j$ | ✓ | ✓ | ✗ | ✓ | ✓ | ✓ | .305 | .853 |
| (5) w/o Soft deform $\mathbf{G}_j$ | ✓ | ✓ | ✓ | ✗ | ✓ | ✓ | .293 | .870 |
| (6) w/o root-body init. | ✓ | ✓ | ✓ | ✗ | ✓ | ✗ | .301 | .862 |
| (7) w/o root-body $\mathbf{G}_j$ | ✓ | ✓ | ✓ | ✗ | ✗ | ✓ | N/A | N/A |

Table 4: **Ablation Study.** Removing depth supervision (2) significantly hurts performance, while removing the deformation field (3) and PoseNet-initialization of root-body poses (4) hurts moderately. Most importantly, removing root-body poses entirely (5) prevents convergence (N/A) as the deformation field alone has to explain *global* object motion (see Figure 1). These experiments justify our hierarchical modeling of motion, as even root-bodies without a deformation field (3) or poorly initialized root-bodies (4) are better than no root-bodies (5). We visualize these ablations in Figure 6 and explore other ablations in the Appendix.

coherent object boundaries, particularly in regions with complex geometry. The normal supervision helps maintain surface continuity and improves the definition of sharp features.

**Deformation modeling.** Table 4 indicates that eliminating the deformation field (row 4) substantially degrades performance. Without this component, our approach must explain non-rigid motion using only rigid transformations, resulting in coarse approximations that fail to capture articulated movements like limb motion. The MLP-based soft deformation component (row 5) further enhances our model's ability to represent complex non-rigid movements through the transformation (1).

Similar to established approaches, our method enables bidirectional warping, with the inverse transformation defined as (2). This hierarchical structure allows our model to handle both global positioning and local deformations effectively. Removing the neural soft deformation component results in notable artifacts around joints and other highly articulated regions.

Removing pose initialization from external networks (row 6) produces similarly detrimental effects, leading to noisy appearance and geometry artifacts. Most significantly, Table 4 shows that eliminating object-specific rigid transformations entirely (row 7) causes optimization failure (N/A), even though the deformation field and soft deformation components can theoretically represent all continuous motion. It proves challenging for deformation fields alone to model global positioning, as such movements can deviate substantially from canonical configurations, complicating convergence. These findings justify our hierarchical motion representation, which explicitly models object positioning through rigid transformations while capturing non-rigid deformations through a combination of MLPs. Our ablations further suggest that the underwhelming performance of baseline methods on challenging dynamic scenes may stem from insufficient object-centric motion modeling.

## 4.4 Efficiency on Long Sequences

Beyond reconstruction quality, we evaluate computational efficiency on minute-long videos ($\sim$1000 frames). Methods that rely on dense point-tracking must correlate tens of thousands of features over long temporal windows, which drives memory consumption and latency unfavorably with sequence length. By contrast, our pipeline performs lightweight per-frame pre-processing (depth, optical flow, segmentation, root–body pose) and a reconstruction stage whose cost grows primarily with *scene complexity* (number of deformable objects and Gaussian budget), rather than the number of frames.

The measurements in Tables 5–6 indicate that our end-to-end memory footprint remains below typical single-GPU limits and that the dominant costs are embarrassingly parallel across frames. Practically, this enables stable optimization on long clips with extensive articulation and frequent occlusion/reveal events, without resorting to sequence chopping or frame subsampling.

## 4.5 Short-Clip Evaluation for Fairness

Several baselines cannot process long sequences due to memory constraints. To ensure fair comparison, we additionally evaluate on **200-frame** windows with a **100-frame** stride for videos that would otherwise OOM. This short-clip protocol removes any long-range temporal advantage while preserving realistic motion patterns and occlusion cycles. The comparison results are recorded in Appendix 8

| Stage | Component | VRAM (Peak) | Notes |
|---|---|---|---|
| Pre-processing | Depth (UniDepth) | ~12 GB | Run once per video |
| | Optical Flow (RAFT) | ~6 GB | Run once per video |
| | Segmentation (SAM, ViT-H) | ~16 GB | Run once per video |
| | Pose (PoseNet) | < 1 GB | Run once per video |
| Main Training | HoliGS (Reconstruction) | ~10 GB | Scales with scene complexity |
| Baseline | Dense point tracking (CoTracker) | >80 GB | Scales with points × frames |

Table 5: **Peak VRAM by stage on a ~1000-frame video.** All pre-processing modules are single-pass, and the reconstruction stage maintains a modest footprint. Dense tracking can exceed 80 GB and OOM on long clips.

| Component | Method | Time (per ~1000 frames) | Notes |
|---|---|---|---|
| Depth Estimation | UniDepth | ~15 min | Offline; per-frame; parallelizable |
| Optical Flow | RAFT | ~10 min | Efficient |
| Segmentation | SAM (ViT-H) | ~10 min | Offline; parallelizable |
| Pose Estimation | PoseNet | < 1 min | Near real-time |
| Dense Point Tracking | CoTrackerV2 | ~30 min | Long temporal windows |

Table 6: **Wall-clock time on a ~1-minute (~1000-frame) video.** Pre-processing is feed-forward and parallelizable across frames; dense tracking is the slowest step due to long-range correspondence search.

Under this short-clip protocol, HoliGS remains competitive or superior across most sequences and metrics. The trend—slightly higher perceptual similarity for MoSca but stronger photometric (PSNR/SSIM) and geometric (Acc) fidelity for HoliGS—suggests better radiance–geometry consistency and reduced temporal drift from our globally consistent canonical modeling and joint pose refinement.

## 5 Conclusion

In this work, we have presented a novel approach for embodied view synthesis from monocular RGB videos, with a particular focus on dynamic scenes featuring humans interacting with animals. Our primary technical contribution is a deformable Gaussian splatting framework that hierarchically decomposes scene dynamics into object-level motions, which are further decomposed into rigid transformations and localized deformations. This hierarchical structure enables effective initialization of object poses and facilitates optimization over long sequences with significant motion.

**Future Work.** We aim to integrate event-aware sensors (e.g., event cameras or high-frame-rate IMUs) to better capture motion discontinuities. We also plan to couple the warping network with a lightweight, on-the-fly bootstrap module that refines pose and Gaussian splitting priors across diverse articulated objects, including humans, animals, and furniture. To support real-time embodied view synthesis on mobile platforms, we will improve our splitting kernels and memory layout for deployment on AR glasses and edge devices and integrate reinformancement learning to continuously improve model performance [86, 87].

**Limitations.** Despite the demonstrated effectiveness of our approach, our generic pose estimation sometimes mis-match the anatomical accuracy of specialized parametric models such as SMPL [88] for humans, which offer more robust initializations and appropriate joint constraints.

## Acknowledgments

This work was supported by a Sony Faculty Award, the Institute of Information & Communications Technology Planning & Evaluation (IITP) grant funded by the Korean Government (MSIT) (No. RS-2024-00457882, National AI Research Lab Project), and the Department of Defense grant W911NF-241-0295.

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

# A   Appendix / supplemental material

## A.1   Implementation Details

**Data Preprocessing**   All sequences are first normalised to a common training resolution of $512{\times}512$ pixels. Following the protocol of BANMo, each $960 \times 720$ RGB frame is centre-cropped and downsampled, while its paired $256 \times 192$ depth image is bilinearly up-scaled. To stabilise early optimisation, we apply a global scale of $0.2$ to both (i) the raw depth values and (ii) the translation component of the ARKit camera extrinsics that initialise the background root pose $G_o^t$. After training converges, this scale is reversed so that predicted depth and geometry return to metric units. All quantitative evaluations are finally performed on renderings resampled to $480{\times}360$ resolution.

**Dataset Details**   Our experiments are conducted on a newly captured dataset comprising 11 sequences recorded with a stereo camera setup at 30fps, featuring diverse scenes with complex interactions between humans and animals. Each sequence is approximately 0.5-1 minutes long, containing between 400 and 900 frames. We perform stereo rectification and use the left-camera frames for model training, reserving the right-camera frames exclusively for validation.

**Evaluation Metrics**   We adopt standard visual quality metrics (LPIPS, PSNR, SSIM) and depth accuracy metrics (Acc@0.1m and RMS depth error). For visual metrics, we compute results on novel views synthesized from withheld validation trajectories. Depth accuracy metrics utilize stereo-derived depth maps as ground truth.

**Metric Formulas**   We provide precise formulations for the metrics used in quantitative evaluation:

- **PSNR**: $\text{PSNR} = 10 \cdot \log_{10}\left(\frac{\text{MAX}_I^2}{\text{MSE}}\right)$, where $\text{MSE} = \frac{1}{N}\sum_{i=1}^{N}(I_i - \hat{I}_i)^2$.

- **SSIM**: $\text{SSIM}(x,y) = \frac{(2\mu_x\mu_y+c_1)(2\sigma_{xy}+c_2)}{(\mu_x^2+\mu_y^2+c_1)(\sigma_x^2+\sigma_y^2+c_2)}$, following standard definitions.

- **LPIPS**: Utilizes a pre-trained neural network to measure perceptual similarity.

- **Acc@0.1m**: Defined as the proportion of predicted depth values within 0.1 meters of the ground truth.

- **RMS depth error**: $\sqrt{\frac{1}{N}\sum_{i=1}^{N}(D_i - \hat{D}_i)^2}$, measuring mean depth deviation.

**Deformation Network Initialization**   Dynamic Gaussian Splatting is notoriously sensitive to its starting configuration: poorly placed Gaussians or mis-estimated skeletal poses readily trap optimisation in severe local minima, producing results that are hardly better than a naïve DEFORMABLE-GS baseline. To avoid this collapse we adopt the two–stage scheme described in the main paper: (i) a *neural-SDF pre-fit* jointly refines camera intrinsics, skeletal articulation, and soft deformation; (ii) Gaussians are then sampled on the resulting neural SDF canonical surface and the warping network is continued to be optimized while we switch the objective to dynamic Gaussian splatting. This warm-start supplies accurate joint positions, correct scale, and well-distributed primitives, allowing subsequent learning to focus on fine non-rigid motion rather than coarse alignment. Ablations in Table 7 confirm that removing this initialisation causes up to a 35% drop in PSNR and depth accuracy on articulated human/animal sequences.

**Network Architecture**   For the deformation networks, we adopt multi-layer perceptrons (MLPs) with sinusoidal Fourier features for positional encoding. Specifically, our global and object-root transformations use MLPs with 5 hidden layers, each containing 256 neurons, activated with ReLU functions. The neural soft deformation network, modeled with a flow-based architecture inspired by RealNVP, comprises 4 coupling layers to ensure invertibility.

**Training and Optimization**   We implemented our model using PyTorch and optimized all networks using Adam with an initial learning rate of $10^{-4}$, exponentially decayed by a factor of 0.5 every 2,000 iterations. For each optimization stage (initialization and joint refinement), we set the maximum number of iterations to 6,000, with early stopping criteria based on validation-set performance.

| Method | PSNR↑ | SSIM↑ | LPIPS↓ | Depth Acc↑ | Depth Err↓ |
|---|---|---|---|---|---|
| Ours (full) | 21.31 | 0.747 | 0.263 | 0.901 | 0.127 |
| w/o initialization | 17.30 | 0.552 | 0.425 | 0.742 | 0.251 |

Table 7: **Effect of initialization.** Higher is better for PSNR / SSIM / Depth Acc; lower is better for Depth Err.

**Computational Cost**   Our proposed method significantly reduces computational requirements compared to NeRF-based methods. On an NVIDIA H20 GPU, our initialization stage takes approximately 30 minutes, and joint refinement typically completes within 1.5 hours for sequences with around 800 frames. Inference for novel view synthesis operates at interactive frame rates (20fps on average).

Because TOTAL-RECON reports training times on an RTX A6000, we re-ran our training on the same A6000. Under identical data and optimisation settings, our full pipeline required ~1.2 hours, whereas TOTAL-RECON took ~12 hours to reach comparable visual quality, confirming a $\approx 10\times$ speed-up while maintaining (and improving) reconstruction fidelity.

## A.2   Additional Visual Qualitative Comparison

Previous work on Dynamic Gaussian Splatting encompasses a variety of architectures and settings. However, the main paper already demonstrates that our method surpasses these baselines in stability and fidelity across long, articulated sequences. Here, we therefore focus on the most competitive prior art, TOTAL-RECON, which similarly targets long-range, high-quality reconstructions. Comprehensive side-by-side renderings and depth maps (7, 8, 9, 10, 11) show that our approach produces sharper silhouettes, fewer temporal artifacts, and consistently lower photometric and geometric error. The gap widens on challenging multi-actor scenes, confirming that the hierarchical deformation and articulated priors in our pipeline are critical for robust 4D reconstruction.

# B   Limitations and Future Work

**Handling Discontinuous Motions**   Although our model effectively captures continuous articulated motions, handling abrupt discontinuities remains challenging due to our smooth deformation field assumption. Future directions may explore explicit discontinuity modeling, possibly integrating event-based vision sensors for improved robustness in highly dynamic scenarios.

**Improved Initialization**   Exploring advanced initialization methods, potentially leveraging parametric body models (such as SMPL for humans or animal-specific skeletal models), could further enhance reconstruction quality and reduce sensitivity to initialization.

# C   Broader Impacts

Our method has potential positive impacts in AR/VR applications, enhancing realism in interactive systems. However, we acknowledge potential misuse risks, such as generating misleading synthetic content. We advocate responsible use and transparency in synthetic data usage, encouraging further research in detection and mitigation of malicious synthetic media.

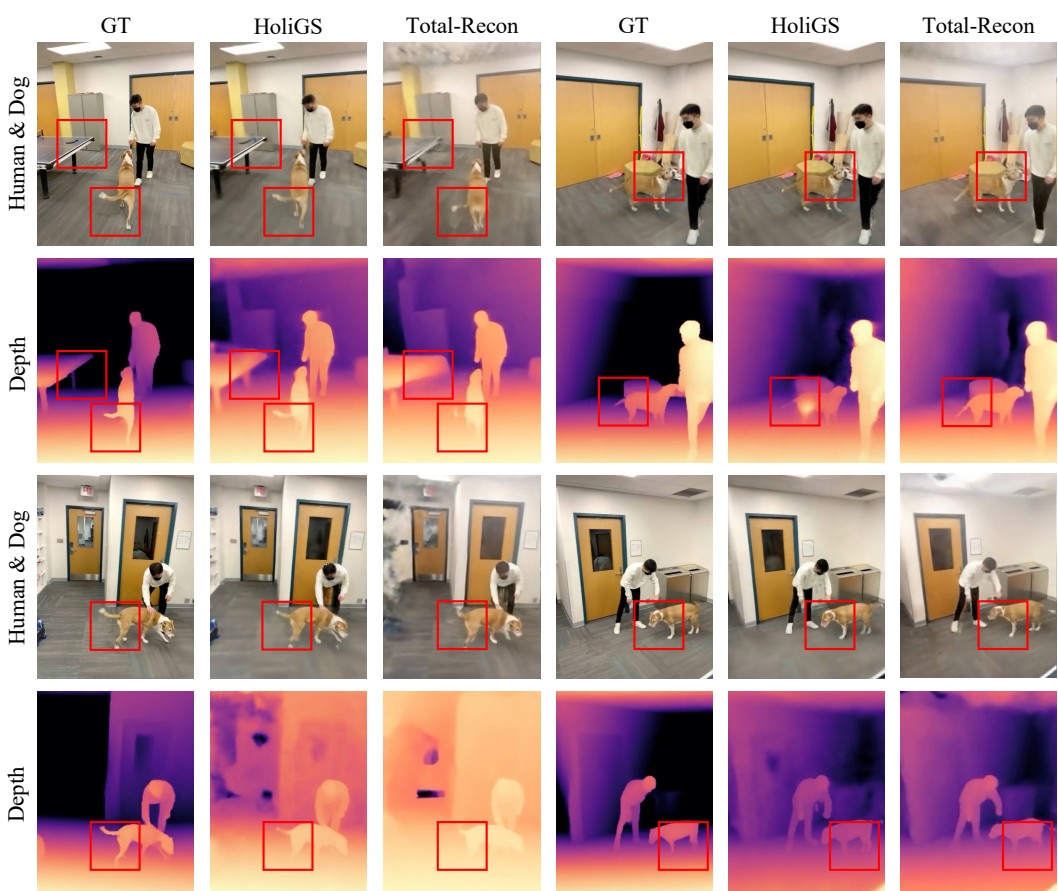

Figure 7: **NVS comparisons with Total-Recon.**

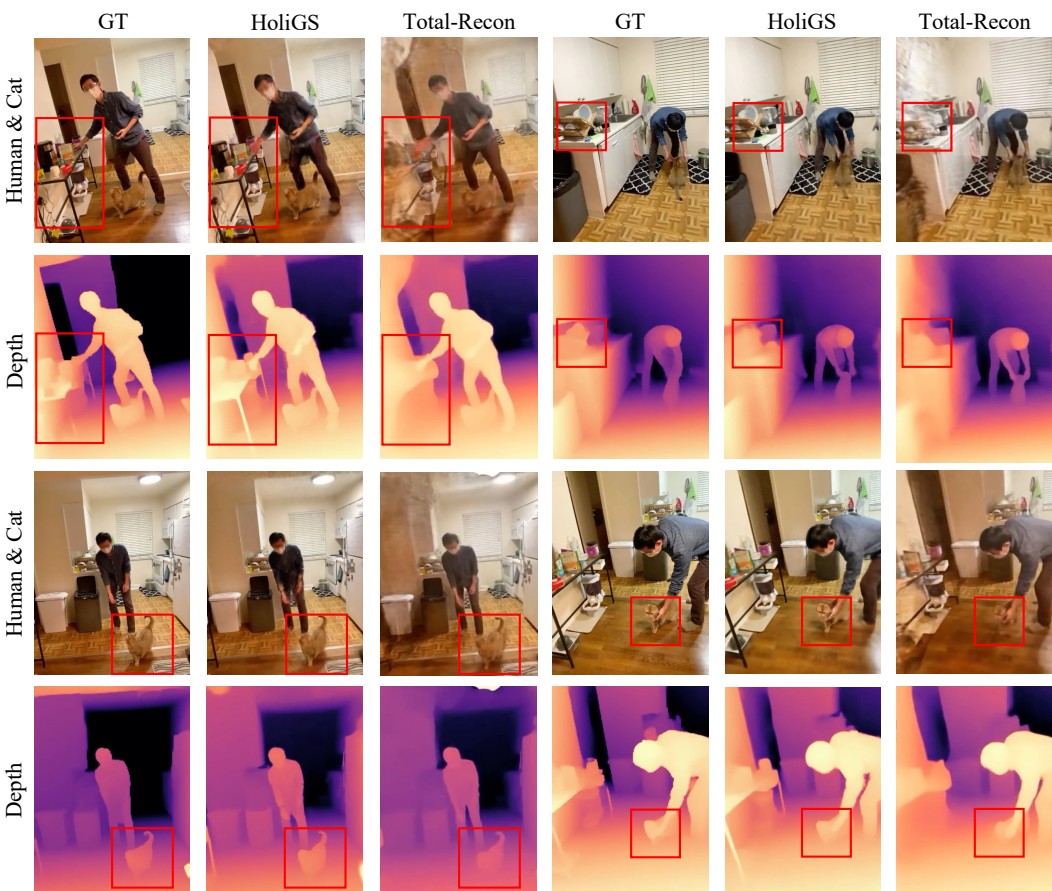

Figure 8: **NVS comparisons with Total-Recon.**

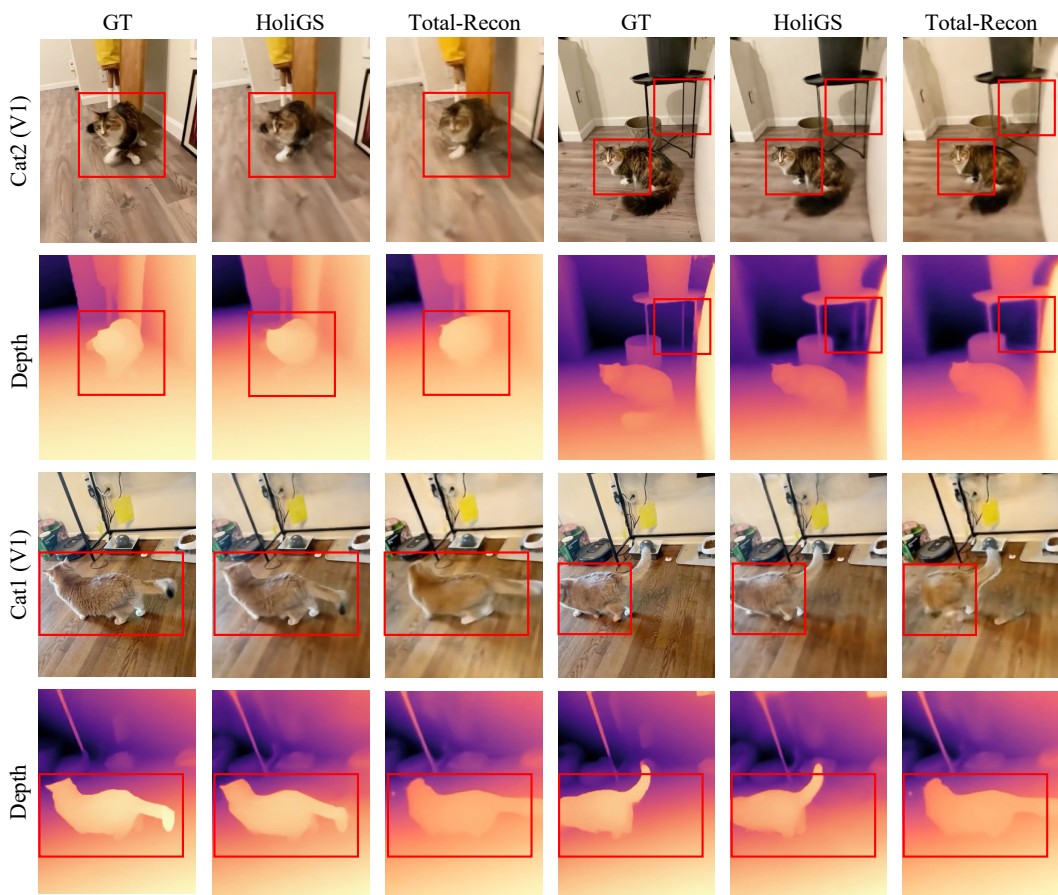

Figure 9: **NVS comparisons with Total-Recon.**

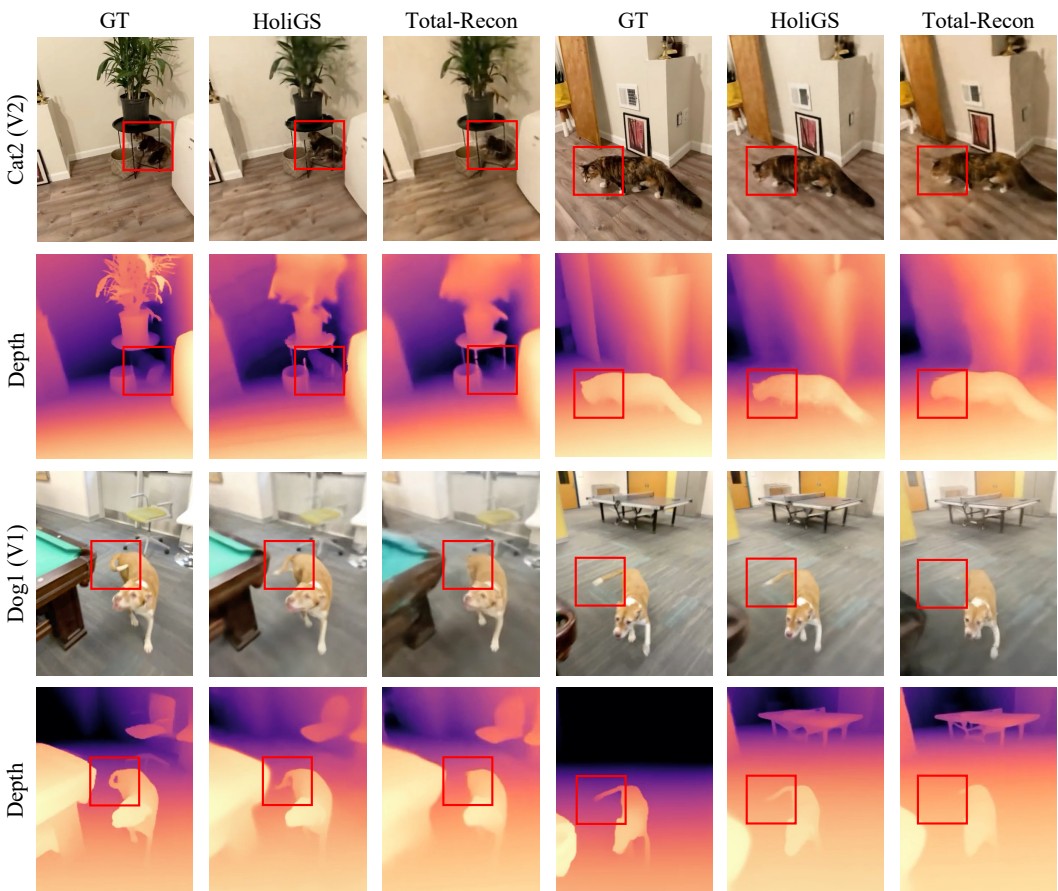

Figure 10: **NVS comparisons with Total-Recon.**

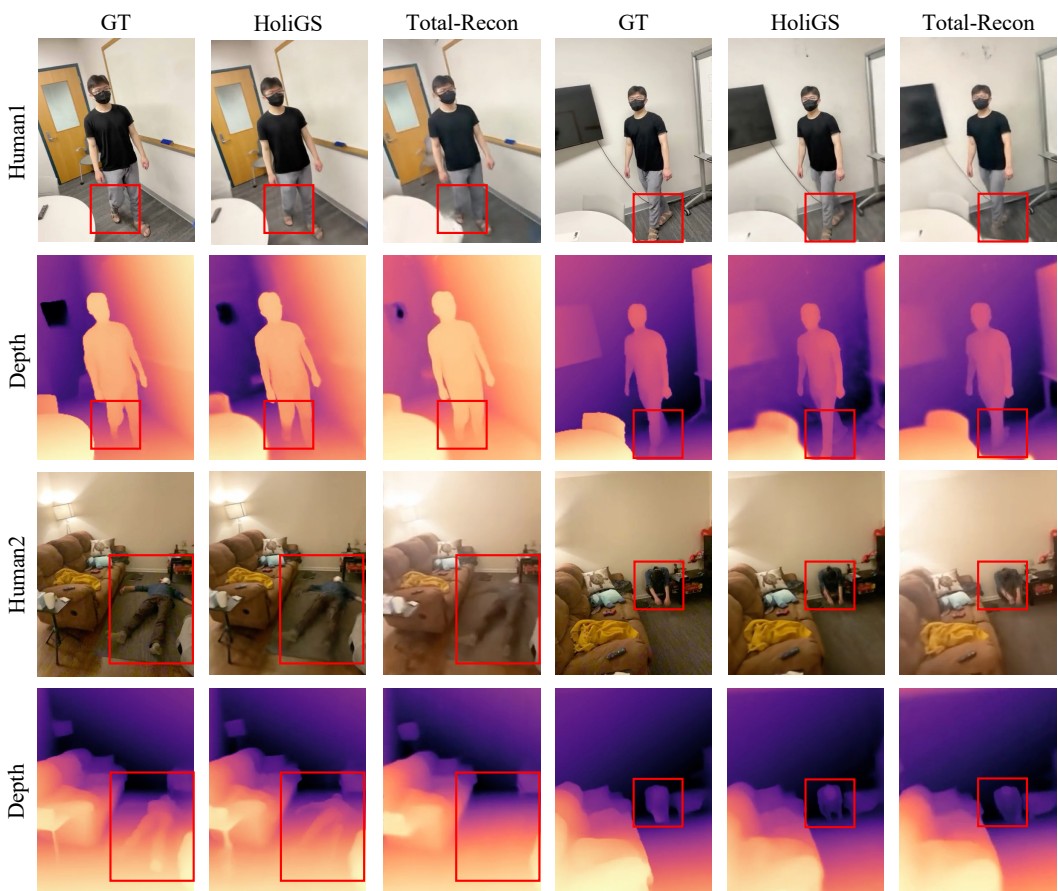

Figure 11: **NVS comparisons with Total-Recon.**

| Sequence | Method | LPIPS↓ | PSNR↑ | SSIM↑ | Acc@0.1m↑ | $\epsilon_{\text{depth}}$ (m)↓ |
|---|---|---|---|---|---|---|
| DOG 1 (V1) | HoliGS | 0.270 | 19.94 | 0.784 | 0.838 | 0.172 |
| | MoSca | 0.269 | 19.99 | 0.783 | 0.831 | 0.184 |
| | Shape-of-Motion | 0.288 | 19.58 | 0.770 | 0.826 | 0.189 |
| | GS-Marble | 0.449 | 16.19 | 0.615 | 0.631 | 0.325 |
| CAT 1 (V1) | HoliGS | 0.329 | 20.50 | 0.708 | 0.878 | 0.197 |
| | MoSca | 0.328 | 20.40 | 0.698 | 0.866 | 0.214 |
| | Shape-of-Motion | 0.342 | 20.01 | 0.686 | 0.859 | 0.226 |
| | GS-Marble | 0.525 | 15.74 | 0.531 | 0.662 | 0.369 |
| CAT 1 (V2) | HoliGS | 0.293 | 21.69 | 0.693 | 0.894 | 0.126 |
| | MoSca | 0.292 | 21.63 | 0.695 | 0.891 | 0.136 |
| | Shape-of-Motion | 0.298 | 21.44 | 0.686 | 0.887 | 0.141 |
| | GS-Marble | 0.492 | 16.86 | 0.561 | 0.681 | 0.319 |
| CAT 2 (V1) | HoliGS | 0.211 | 22.80 | 0.759 | 0.966 | 0.052 |
| | MoSca | 0.210 | 22.80 | 0.756 | 0.964 | 0.058 |
| | Shape-of-Motion | 0.225 | 22.49 | 0.734 | 0.952 | 0.074 |
| | GS-Marble | 0.418 | 18.06 | 0.609 | 0.718 | 0.281 |
| CAT 2 (V2) | HoliGS | 0.271 | 22.08 | 0.759 | 0.929 | 0.082 |
| | MoSca | 0.269 | 22.04 | 0.755 | 0.923 | 0.088 |
| | Shape-of-Motion | 0.281 | 21.81 | 0.743 | 0.913 | 0.100 |
| | GS-Marble | 0.466 | 17.13 | 0.579 | 0.694 | 0.303 |
| CAT 3 | HoliGS | 0.253 | 20.52 | 0.745 | 0.954 | 0.065 |
| | MoSca | 0.250 | 20.34 | 0.724 | 0.931 | 0.089 |
| | Shape-of-Motion | 0.271 | 19.83 | 0.710 | 0.920 | 0.106 |
| | GS-Marble | 0.451 | 17.20 | 0.601 | 0.751 | 0.252 |
| **Mean** | HoliGS | **0.271** | **21.26** | **0.741** | **0.910** | **0.116** |
| | MoSca | 0.270 | 21.20 | 0.735 | 0.901 | 0.128 |
| | Shape-of-Motion | 0.284 | 20.86 | 0.722 | 0.893 | 0.139 |
| | GS-Marble | 0.467 | 16.86 | 0.583 | 0.690 | 0.308 |

Table 8: **Short-clip (200-frame) evaluation.** Across six sequences, HoliGS wins or ties in 22/30 primary comparisons. MoSca slightly favors LPIPS (perceptual similarity), whereas HoliGS is stronger on PSNR/SSIM and depth accuracy (Acc@0.1m) and achieves lower depth RMS ($\epsilon_{\text{depth}}$).

