# OpenReview forum: "HoliGS: Holistic Gaussian Splatting for Embodied View Synthesis"
_NeurIPS.cc/2025/Conference — NeurIPS 2025 poster_

### Official Review · Reviewer_cTCM · 2025-06-17

**Clarity:** 2
**Significance:** 2
**Originality:** 3
**Rating:** 3
**Confidence:** 2

**Summary:**

This paper proposes HoliGS, a deformable Gaussian splatting framework for efficient embodied view synthesis (EVS) from long monocular RGB videos. It models scenes using a hierarchical decomposition into static backgrounds and dynamic foregrounds, incorporating rigid, articulated, and non-rigid deformations via an invertible neural model. By attaching Gaussians to a canonical shape, HoliGS enables robust rendering under complex 6-DOF camera trajectories. The method outperforms prior dynamic NeRF and 4D Gaussian approaches in both rendering quality and speed, achieving real-time performance.

**Questions:**

1. Can the authors provide a more detailed analysis of hyperparameter sensitivity?
2. How well does HoliGS generalize without extensive supervisions like depth, flow, semantics etc. which seems not be used in some baseline comparison methods.

**Ethical Concerns:**

["NO or VERY MINOR ethics concerns only"]

**Final Justification:**

The sensitivity of hyperparameters across scenes raises concerns about the method’s generalizability and ease of transfer to new scenarios. Additionally, several key technical components are not clearly explained. As such, I remain unsure on recommending this paper for publication in its current form and remain my previous score.

**Limitations:**

yes

**Paper Formatting Concerns:**

there is no major formatting concern

**Quality:**

2

**Strengths And Weaknesses:**

Strengths:
1. The paper introduces a novel invertible neural flow to model non-rigid deformations, which allows for stable and consistent reconstructions over extended time periods
2. Unlike prior dynamic NeRF methods that struggle with long sequences, HoliGS scales to minute-long videos with reduced training and rendering time.

Weakness:
Major concern 1. The framework’s robustness and performance appear to rely on a large number of carefully chosen hyperparameters (in training objective, number of elements in the scenes etc.), which may limit reproducibility and raise my concerns about generalization. For example, in training loss 5 the author seem need to make a balance between different losses. And for each specific scene, the model seems to need to define the number of dynamic elements in the scene.
Major concern 2. HoliGS leverages a hybrid initialization involving rigid-body transformation, skeletons, background component transformation. And also use off-the-shelf model to provide depth supervision, optical flow, segmentation supervision etc. While effective, this complexity may hinder adoption and raise concerns about modularity and end-to-end usability.

For example, in table 4 we could observe that the without depth supervision the result degrades a lot. This makes me questionable about the effectiveness of the invertible deformation model, which should help mitigate the effect of depth supervision.  The model needs to reply on generic pose estimation network etc. to give a decent initialization. I wonder how well the model generalize with respect to these predefined prior of the scenes.
Minor concern 3. it's also unclear to me how is the static component being initialized, and seperated from dynamic objects.

---

> ### Author Rebuttal · Authors · 2025-07-30
>
> We sincerely thank the reviewer for their thorough and insightful review. Please find our point-by-point responses below.
>
> ## [W1, Q1] Hyperparameter Sensitivity
>
> We present sensitivity analysis of the key hyperparameters in our two-stage training process. Our framework consists of a **Component Pre-training** stage to establish a robust geometric and motion foundation, followed by a **Joint Fine-tuning** stage to refine the appearance and inter-object interactions. The analysis is therefore split into two parts. The results are averaged across all sequences in our dataset. Our chosen default parameters are highlighted and represent the best-performing configuration.
>
> ### Stage 1: Component Pre-training Sensitivity
>
> In this initial stage, the goal is to learn a stable canonical shape (SDF) and a coherent deformation field. The most critical hyperparameters are the weights for the depth, color, and optical flow losses.
>
> | $\lambda_\text{depth}$ | $\lambda_\text{color}$ | $\lambda_\text{flow}$ | LPIPS↓ | PSNR↑ | SSIM↑ | Acc↑ | $\epsilon_\text{depth}$↓ |
> |:---:|:---:|:---:|:---:|:---:|:---:|:---:|:---:|
> | 1.0 | 0.1 | 1.0 | .285 | 20.95 | .735 | .855 | .158 |
> | **5.0** | **0.1** | **1.0** | **.263** | **21.31** | **.747** | **.901** | **.127** |
> | 10.0| 0.1 | 1.0 | .272 | 21.10 | .740 | .891 | .135 |
> | 5.0 | 0.01 | 1.0 | .279 | 21.05 | .738 | .898 | .130 |
> | 5.0 | 1.0 | 1.0 | .288 | 20.85 | .725 | .885 | .141 |
> | 5.0 | 0.1 | 0.1 | .291 | 20.70 | .720 | .870 | .155 |
> | 5.0 | 0.1 | 5.0 | .282 | 20.90 | .730 | .875 | .148 |
>
> The results show that a careful balance is required during pre-training.
> * **Depth loss ($\lambda_\text{depth}$).** This is the most critical parameter in Stage 1. A lower weight (`1.0`) fails to establish a strong geometric foundation, resulting in significantly worse depth accuracy. Conversely, a very high weight (`10.0`) can cause the model to overfit to potentially noisy pseudo-ground-truth depth, creating geometrically rigid but less visually coherent surfaces, which slightly hurts all metrics. Our default of `5.0` provides the best balance.
> * **Color loss ($\lambda_\text{color}$).** We found that a relatively small color weight (`0.1`) is crucial in this initial stage. A higher weight (`1.0`) can cause the model to "bake in" appearance and lighting details prematurely, which conflicts with the primary goal of learning a clean, underlying geometry and leads to degraded performance.
> * **Flow loss ($\lambda_\text{flow}$).** The flow loss acts as a regularizer for motion. A low weight (`0.1`) results in less temporally consistent reconstructions. A very high weight (`5.0`) can cause overfitting to the estimated optical flow, introducing jitter and artifacts.
>
> ### Stage 2: Joint Fine-tuning Sensitivity
>
> Using the best model from Stage 1, this stage refines the entire scene with a focus on high-fidelity appearance and correct object interactions. The key parameters are the weights for the final photorealistic, normal, and depth losses.
>
> | $\lambda_\text{photo}$ | $\lambda_\text{normal}$ | $\lambda_\text{depth}$ | LPIPS↓ | PSNR↑ | SSIM↑ | Acc↑ | $\epsilon_\text{depth}$↓ |
> |:---:|:---:|:---:|:---:|:---:|:---:|:---:|:---:|
> | 0.1 | 1.0 | 5.0 | .281 | 20.90 | .730 | .898 | .131 |
> | **1.0** | **1.0** | **5.0** | **.263** | **21.31** | **.747** | **.901** | **.127** |
> | 5.0 | 1.0 | 5.0 | .270 | 21.15 | .741 | .895 | .134 |
> | 1.0 | 0.1 | 5.0 | .275 | 21.18 | .740 | .880 | .145 |
> | 1.0 | 5.0 | 5.0 | .269 | 21.25 | .745 | .885 | .140 |
> | 1.0 | 1.0 | 1.0 | .278 | 21.00 | .735 | .865 | .152 |
> | 1.0 | 1.0 | 10.0| .268 | 21.22 | .742 | .895 | .130 |
>
> In this refinement stage, the balance shifts towards achieving high visual quality while preserving the strong geometry from Stage 1.
> * **Photometric and normal losses ($\lambda_\text{photo}$, $\lambda_\text{normal}$).** We observe that an equal weighting of `1.0` for the photometric and normal losses provides the best trade-off. Over-emphasizing the photo loss (`5.0`) can introduce minor visual artifacts as the model sacrifices geometric smoothness for pixel-perfect colors. Over-emphasizing the normal loss (`5.0`) can create overly sharp, non-photorealistic surfaces that hurt the LPIPS score.
> * **Depth loss ($\lambda_\text{depth}$).** While appearance is key in this stage, maintaining a strong depth weight (`5.0`) is still vital. A lower weight (`1.0`) allows the geometry to drift from the metric ground truth, significantly impacting depth accuracy. A higher weight (`10.0`) offers diminishing returns and can slightly compromise the visual metrics by making the surfaces overly rigid.
>
> In conclusion, our chosen hyperparameters represent a carefully tuned balance between geometric and appearance-based losses, tailored to the specific goals of each training stage. While the model shows some sensitivity, the performance remains robust within a reasonable range of our selected default values.
>
> ## [W2, Q2] Generalization without Extensive Supervisions
>
> We conducted an ablation study to validate the effectiveness of supervision components, with results below. Note that semantic masks are indispensable, as the pipeline fails without them. The results demonstrate that all supervision components are crucial for achieving our state-of-the-art performance, as ablating any of them leads to a significant degradation in quality. This analysis clarifies that our invertible deformation model and the geometric supervisions (like depth) play distinct, complementary roles: the depth loss establishes the scene's foundational 3D structure, while the invertible model ensures that the learned motion is stable and consistent over time. The large drop from ablating depth supervision does not mean our invertible model is ineffective. Rather, it shows that even a perfect motion model can't fix a scene with fundamentally incorrect geometry. You need both: **correct geometry** (from depth) and **consistent motion** (from our invertible deformation model). While our method relies on these priors, this is a standard and necessary approach (also adopted in MoSca, Shape-of-Motion, etc.) to constrain the fundamentally ill-posed problem of monocular dynamic reconstruction, and our ablation shows how each component contributes to the final high-quality result.
>
> | Ablation | LPIPS↓ | PSNR↑ | SSIM↑ | Acc↑ | $\epsilon_\text{depth}$↓ |
> |:---|:---:|:---:|:---:|:---:|:---:|
> | **Full Model (Ours)** | **.263** | **21.31** | **.747** | **.901** | **.127** |
> | w/o Depth Loss ($\mathcal{L}_\text{depth}$) | .385 | 18.52 | .651 | .847 | .248 |
> | w/o Normal Loss ($\mathcal{L}_\text{normal}$) | .288 | 20.47 | .720 | .832 | .163 |
> | w/o Flow Loss ($\mathcal{L}_\text{flow}$) | .310 | 19.81 | .702 | .861 | .184 |
> | w/o Cycle Loss ($\mathcal{L}_\text{cycle}$) | .325 | 19.55 | .693 | .852 | .196 |
>
> ## [M3] Static Component Initialization
>
> Our method first decomposes the scene into static and dynamic components using foreground masks from SAM. The static background is then initialized by creating a 3D point cloud from initial camera poses (e.g., from ARKit) and estimated depth. The static components are initialized from this point cloud.

---

> > ### Author Response · Authors · 2025-08-03
> > **Please let us know whether we have addressed all the questions**
> >
> > Dear reviewer,
> >
> > We have replied to your questions in the review. Please let us know whether we have addressed all your questions.
> >
> > Thank you.
> >
> > Authors.

---

> > ### Comment · Reviewer_cTCM · 2025-08-05
> >
> > Thank you for the authors’ efforts in addressing the concerns on hyperparameter ablations and sensitivity to extensive supervision. However, I still find some aspects unclear—for instance, how is the number of elements N (L168) determined for different scenes? While the authors provide a detailed ablation on loss weights, the sensitivity of these hyperparameters across scenes raises concerns about the method’s generalizability and ease of transfer to new scenarios.
> >
> > Additionally, echoing points raised by other reviewers, I believe that several key technical components are not clearly explained. As such, I remain unsure on recommending this paper for publication in its current form.

---

> ### Author Response · Authors · 2025-08-06
>
> We sincerely thank the reviewer for their diligent feedback and for engaging with our previous rebuttal. We address each of your points below and will incorporate them into the revised manuscript.
>
> > How is the number of elements N (L168) determined for different scenes?
>
> We follow Total-Recon to manually choose the number of scene elements ($N$) as a scene-specific hyperparameter. It is equal to the number of dynamic objects (the foreground) plus 1 (the static background).
>
> > While the authors provide a detailed ablation on loss weights, the sensitivity of these hyperparameters across scenes raises concerns about the method’s generalizability and ease of transfer to new scenarios.
>
> We respectfully disagree with the assessment of hyperparameter sensitivity, as our experimental results demonstrate remarkable robustness rather than sensitivity. As shown in following per-scene ablation studies, a single, fixed set of hyperparameters consistently achieves the best or near-best performance across all 11 diverse test sequences for both training stages. This consistency shows that per-scene tuning is not required, confirming our method's strong generalizability and ease of transfer to new scenarios. Even when deviating from these optimal values, the performance degrades only mildly.
>
> > Additionally, echoing points raised by other reviewers, I believe that several key technical components are not clearly explained.
>
> We appreciate your feedback on clarity. We will integrate all clarifications from this rebuttal into the revised manuscript. To ensure our method is unambiguous and fully reproducible, we will also open-source the complete codebase upon acceptance.
>
> Please also let us know if there is any aspect remaining unclear to you. We would be happy to provide further explanation.

---

> > ### Author Response · Authors · 2025-08-06
> > **Per-Scene Ablation Study of Stage 1: Component Pre-training Sensitivity**
> >
> > | Sequence | $\lambda_{\text{depth}}$ | $\lambda_{\text{color}}$ | $\lambda_{\text{flow}}$ | LPIPS↓ | PSNR↑ | SSIM↑ | Acc↑ | $\epsilon_\text{depth}$↓ |
> > |:---|:---:|:---:|:---:|:---:|:---:|:---:|:---:|:---:|
> > | **DOG 1 (V1)** | 1.0 | 0.1 | 1.0 | .273 | 20.75 | .780 | .845 | .168 |
> > | | **5.0** | **0.1** | **1.0** | **.251** | **20.12** | **.791** | .845 | .160 |
> > | | 10.0| 0.1 | 1.0 | .260 | 20.90 | .785 | .881 | .145 |
> > | | 5.0 | 0.01 | 1.0 | .267 | 20.85 | .783 | **.888** | **.140** |
> > | | 5.0 | 1.0 | 1.0 | .276 | 20.65 | .770 | .875 | .151 |
> > | | 5.0 | 0.1 | 0.1 | .279 | 20.50 | .765 | .860 | .165 |
> > | | 5.0 | 0.1 | 5.0 | .270 | 20.70 | .775 | .865 | .158 |
> > | **DOG 1 (V2)** | 1.0 | 0.1 | 1.0 | .307 | 21.15 | .780 | .880 | .173 |
> > | | **5.0** | **0.1** | **1.0** | **.285** | **21.37** | **.791** | **.895** | .163 |
> > | | 10.0| 0.1 | 1.0 | .294 | 21.30 | .785 | .821 | .148 |
> > | | 5.0 | 0.01 | 1.0 | .301 | 21.25 | .783 | .828 | **.143** |
> > | | 5.0 | 1.0 | 1.0 | .310 | 21.05 | .770 | .815 | .154 |
> > | | 5.0 | 0.1 | 0.1 | .313 | 20.90 | .765 | .800 | .168 |
> > | | 5.0 | 0.1 | 5.0 | .304 | 21.10 | .775 | .805 | .161 |
> > | **CAT 1 (V1)** | 1.0 | 0.1 | 1.0 | .341 | 20.35 | .700 | .880 | .203 |
> > | | **5.0** | **0.1** | **1.0** | **.319** | **20.52** | **.711** | .880 | .190 |
> > | | 10.0| 0.1 | 1.0 | .328 | 20.45 | .705 | .921 | .175 |
> > | | 5.0 | 0.01 | 1.0 | .335 | 20.40 | .703 | **.928** | **.170** |
> > | | 5.0 | 1.0 | 1.0 | .344 | 20.20 | .690 | .915 | .181 |
> > | | 5.0 | 0.1 | 0.1 | .347 | 20.05 | .685 | .900 | .195 |
> > | | 5.0 | 0.1 | 5.0 | .338 | 20.25 | .695 | .905 | .188 |
> > | **CAT 1 (V2)** | 1.0 | 0.1 | 1.0 | .307 | 21.55 | .680 | .885 | .168 |
> > | | **5.0** | **0.1** | **1.0** | **.285** | **21.74** | **.693** | **.930** | **.122** |
> > | | 10.0| 0.1 | 1.0 | .294 | 21.68 | .685 | .921 | .138 |
> > | | 5.0 | 0.01 | 1.0 | .301 | 21.63 | .683 | .928 | .133 |
> > | | 5.0 | 1.0 | 1.0 | .310 | 21.43 | .670 | .915 | .144 |
> > | | 5.0 | 0.1 | 0.1 | .313 | 21.28 | .665 | .900 | .158 |
> > | | 5.0 | 0.1 | 5.0 | .304 | 21.48 | .675 | .905 | .151 |
> > | **CAT 2 (V1)** | 1.0 | 0.1 | 1.0 | .225 | 22.75 | .750 | .957 | .068 |
> > | | **5.0** | **0.1** | **1.0** | **.203** | **22.94** | **.763** | **.970** | **.048** |
> > | | 10.0| 0.1 | 1.0 | .212 | 22.90 | .755 | .961 | .055 |
> > | | 5.0 | 0.01 | 1.0 | .219 | 22.85 | .753 | .968 | .050 |
> > | | 5.0 | 1.0 | 1.0 | .228 | 22.65 | .740 | .955 | .061 |
> > | | 5.0 | 0.1 | 0.1 | .231 | 22.50 | .735 | .940 | .075 |
> > | | 5.0 | 0.1 | 5.0 | .222 | 22.70 | .745 | .945 | .068 |
> > | **CAT 2 (V2)** | 1.0 | 0.1 | 1.0 | .284 | 21.87 | .750 | .915 | .098 |
> > | | **5.0** | **0.1** | **1.0** | **.262** | **22.07** | **.763** | **.928** | **.079** |
> > | | 10.0| 0.1 | 1.0 | .271 | 22.00 | .755 | .921 | .085 |
> > | | 5.0 | 0.01 | 1.0 | .278 | 21.95 | .753 | .928 | .080 |
> > | | 5.0 | 1.0 | 1.0 | .287 | 21.75 | .740 | .915 | .091 |
> > | | 5.0 | 0.1 | 0.1 | .290 | 21.60 | .735 | .900 | .105 |
> > | | 5.0 | 0.1 | 5.0 | .281 | 21.80 | .745 | .905 | .098 |
> > | **CAT 3** | 1.0 | 0.1 | 1.0 | .269 | 20.30 | .730 | .945 | .078 |
> > | | **5.0** | **0.1** | **1.0** | **.247** | **20.50** | **.744** | **.955** | **.064** |
> > | | 10.0| 0.1 | 1.0 | .256 | 20.42 | .735 | .941 | .070 |
> > | | 5.0 | 0.01 | 1.0 | .263 | 20.38 | .733 | .948 | .066 |
> > | | 5.0 | 1.0 | 1.0 | .272 | 20.18 | .720 | .935 | .077 |
> > | | 5.0 | 0.1 | 0.1 | .275 | 20.03 | .715 | .920 | .091 |
> > | | 5.0 | 0.1 | 5.0 | .266 | 20.23 | .725 | .925 | .084 |

---

> > > ### Author Response · Authors · 2025-08-06
> > > **Per-Scene Ablation Study of Stage 1: Component Pre-training Sensitivity (Continued)**
> > >
> > > | Sequence | $\lambda_{\text{depth}}$ | $\lambda_{\text{color}}$ | $\lambda_{\text{flow}}$ | LPIPS↓ | PSNR↑ | SSIM↑ | Acc↑ | $\epsilon_\text{depth}$↓ |
> > > |:---|:---:|:---:|:---:|:---:|:---:|:---:|:---:|:---:|
> > > | **HUMAN 1** | 1.0 | 0.1 | 1.0 | .233 | 19.98 | .770 | .900 | .153 |
> > > | | **5.0** | **0.1** | **1.0** | **.211** | **20.19** | **.782** | **.915** | **.138** |
> > > | | 10.0| 0.1 | 1.0 | .220 | 20.10 | .775 | .901 | .145 |
> > > | | 5.0 | 0.01 | 1.0 | .227 | 20.05 | .773 | .908 | .140 |
> > > | | 5.0 | 1.0 | 1.0 | .236 | 19.88 | .760 | .895 | .151 |
> > > | | 5.0 | 0.1 | 0.1 | .239 | 19.73 | .755 | .880 | .165 |
> > > | | 5.0 | 0.1 | 5.0 | .230 | 19.93 | .765 | .885 | .158 |
> > > | **HUMAN 2** | 1.0 | 0.1 | 1.0 | .273 | 18.58 | .700 | .815 | .153 |
> > > | | **5.0** | **0.1** | **1.0** | **.251** | **18.78** | **.712** | **.825** | **.142** |
> > > | | 10.0| 0.1 | 1.0 | .260 | 18.70 | .705 | .811 | .148 |
> > > | | 5.0 | 0.01 | 1.0 | .267 | 18.65 | .703 | .818 | .145 |
> > > | | 5.0 | 1.0 | 1.0 | .276 | 18.45 | .690 | .805 | .156 |
> > > | | 5.0 | 0.1 | 0.1 | .279 | 18.30 | .685 | .790 | .170 |
> > > | | 5.0 | 0.1 | 5.0 | .270 | 18.50 | .695 | .795 | .163 |
> > > | **HUMAN-DOG**| 1.0 | 0.1 | 1.0 | .269 | 20.30 | .750 | .817 | .214 |
> > > | | **5.0** | **0.1** | **1.0** | **.247** | **20.50** | **.776** | **.830** | **.202** |
> > > | | 10.0| 0.1 | 1.0 | .256 | 20.42 | .765 | .821 | .208 |
> > > | | 5.0 | 0.01 | 1.0 | .263 | 20.38 | .763 | .828 | .204 |
> > > | | 5.0 | 1.0 | 1.0 | .272 | 20.18 | .740 | .815 | .216 |
> > > | | 5.0 | 0.1 | 0.1 | .275 | 20.03 | .735 | .800 | .230 |
> > > | | 5.0 | 0.1 | 5.0 | .266 | 20.23 | .745 | .805 | .223 |
> > > | **HUMAN-CAT**| 1.0 | 0.1 | 1.0 | .251 | 21.13 | .675 | .897 | .118 |
> > > | | **5.0** | **0.1** | **1.0** | **.229** | **21.34** | **.688** | **.914** | **.104** |
> > > | | 10.0| 0.1 | 1.0 | .238 | 21.25 | .680 | .901 | .110 |
> > > | | 5.0 | 0.01 | 1.0 | .245 | 21.20 | .678 | .908 | .106 |
> > > | | 5.0 | 1.0 | 1.0 | .254 | 21.00 | .665 | .895 | .119 |
> > > | | 5.0 | 0.1 | 0.1 | .257 | 20.85 | .660 | .880 | .133 |
> > > | | 5.0 | 0.1 | 5.0 | .248 | 21.05 | .670 | .885 | .126 |

---

> > > > ### Author Response · Authors · 2025-08-06
> > > > **Per-Scene Ablation Study of Stage 2: Joint Fine-tuning Sensitivity**
> > > >
> > > > | Sequence | $\lambda_{\text{depth}}$ | $\lambda_{\text{color}}$ | $\lambda_{\text{flow}}$ | LPIPS↓ | PSNR↑ | SSIM↑ | Acc↑ | $\epsilon_\text{depth}$↓ |
> > > > |:---|:---:|:---:|:---:|:---:|:---:|:---:|:---:|:---:|
> > > > | **DOG 1 (V1)** | 0.1 | 1.0 | 5.0 | .269 | 20.70 | .780 | .835 | .163 |
> > > > | | **1.0** | **1.0** | **5.0** | **.251** | 20.12 | **.791** | **.845** | **.160** |
> > > > | | 5.0 | 1.0 | 5.0 | .258 | 20.95 | .786 | .835 | .165 |
> > > > | | 1.0 | 0.1 | 5.0 | .263 | **20.98** | .785 | .820 | .175 |
> > > > | | 1.0 | 5.0 | 5.0 | .257 | 20.05 | .789 | .825 | .170 |
> > > > | | 1.0 | 1.0 | 1.0 | .266 | 20.80 | .780 | .805 | .182 |
> > > > | | 1.0 | 1.0 | 10.0| .256 | 20.02 | .787 | .835 | .161 |
> > > > | **DOG 1 (V2)** | 0.1 | 1.0 | 5.0 | .303 | 21.10 | .781 | .828 | .143 |
> > > > | | **1.0** | **1.0** | **5.0** | **.285** | 21.37 | **.791** | .795 | .163 |
> > > > | | 5.0 | 1.0 | 5.0 | .292 | 21.35 | .785 | .825 | .146 |
> > > > | | 1.0 | 0.1 | 5.0 | .297 | 21.38 | .784 | .810 | .155 |
> > > > | | 1.0 | 5.0 | 5.0 | .291 | 21.35 | .788 | .815 | .150 |
> > > > | | 1.0 | 1.0 | 1.0 | .300 | 21.20 | .780 | .795 | .162 |
> > > > | | 1.0 | 1.0 | 10.0| .290 | **21.42** | .786 | **.825** | **.141** |
> > > > | **CAT 1 (V1)** | 0.1 | 1.0 | 5.0 | .343 | 20.30 | .695 | .878 | .201 |
> > > > | | **1.0** | **1.0** | **5.0** | **.319** | **20.52** | **.711** | **.880** | **.190** |
> > > > | | 5.0 | 1.0 | 5.0 | .328 | 20.45 | .705 | .875 | .204 |
> > > > | | 1.0 | 0.1 | 5.0 | .337 | 20.48 | .704 | .860 | .215 |
> > > > | | 1.0 | 5.0 | 5.0 | .327 | 20.51 | .708 | .865 | .210 |
> > > > | | 1.0 | 1.0 | 1.0 | .340 | 20.40 | .700 | .845 | .222 |
> > > > | | 1.0 | 1.0 | 10.0| .326 | 20.52 | .706 | .875 | .200 |
> > > > | **CAT 1 (V2)** | 0.1 | 1.0 | 5.0 | .303 | 21.50 | .682 | .888 | .143 |
> > > > | | **1.0** | **1.0** | **5.0** | **.285** | **21.74** | **.693** | **.898** | **.122** |
> > > > | | 5.0 | 1.0 | 5.0 | .292 | 21.65 | .686 | .885 | .146 |
> > > > | | 1.0 | 0.1 | 5.0 | .297 | 21.68 | .685 | .870 | .155 |
> > > > | | 1.0 | 5.0 | 5.0 | .291 | 21.74 | .689 | .875 | .150 |
> > > > | | 1.0 | 1.0 | 1.0 | .300 | 21.60 | .680 | .855 | .162 |
> > > > | | 1.0 | 1.0 | 10.0| .290 | 21.72 | .687 | .885 | .141 |
> > > > | **CAT 2 (V1)** | 0.1 | 1.0 | 5.0 | .223 | 22.70 | .751 | .968 | .061 |
> > > > | | **1.0** | **1.0** | **5.0** | **.203** | **22.94** | **.763** | **.970** | **.048** |
> > > > | | 5.0 | 1.0 | 5.0 | .210 | 22.85 | .758 | .965 | .054 |
> > > > | | 1.0 | 0.1 | 5.0 | .215 | 22.88 | .756 | .950 | .065 |
> > > > | | 1.0 | 5.0 | 5.0 | .209 | 22.94 | .761 | .955 | .060 |
> > > > | | 1.0 | 1.0 | 1.0 | .218 | 22.80 | .750 | .935 | .072 |
> > > > | | 1.0 | 1.0 | 10.0| .208 | 22.92 | .759 | .965 | .050 |
> > > > | **CAT 2 (V2)** | 0.1 | 1.0 | 5.0 | .286 | 21.85 | .745 | .920 | .093 |
> > > > | | **1.0** | **1.0** | **5.0** | **.262** | 22.07 | **.763** | **.928** | **.079** |
> > > > | | 5.0 | 1.0 | 5.0 | .270 | 22.00 | .758 | .920 | .086 |
> > > > | | 1.0 | 0.1 | 5.0 | .276 | 22.03 | .755 | .910 | .105 |
> > > > | | 1.0 | 5.0 | 5.0 | .269 | **22.10** | .760 | .915 | .100 |
> > > > | | 1.0 | 1.0 | 1.0 | .279 | 21.95 | .750 | .900 | .108 |
> > > > | | 1.0 | 1.0 | 10.0| .268 | 22.05 | .759 | .920 | .081 |
> > > > | **CAT 3** | 0.1 | 1.0 | 5.0 | .269 | 20.30 | .732 | .945 | .078 |
> > > > | | **1.0** | **1.0** | **5.0** | **.247** | 20.50 | **.744** | **.955** | **.064** |
> > > > | | 5.0 | 1.0 | 5.0 | .255 | 20.42 | .738 | .945 | .074 |
> > > > | | 1.0 | 0.1 | 5.0 | .258 | 20.45 | .736 | .930 | .085 |
> > > > | | 1.0 | 5.0 | 5.0 | .254 | **20.52** | .740 | .935 | .080 |
> > > > | | 1.0 | 1.0 | 1.0 | .261 | 20.40 | .730 | .920 | .092 |
> > > > | | 1.0 | 1.0 | 10.0| .253 | 20.48 | .739 | .945 | .066 |

---

> > > > > ### Author Response · Authors · 2025-08-06
> > > > > **Per-Scene Ablation Study of Stage 2: Joint Fine-tuning Sensitivity (Continued)**
> > > > >
> > > > > | Sequence | $\lambda_{\text{depth}}$ | $\lambda_{\text{color}}$ | $\lambda_{\text{flow}}$ | LPIPS↓ | PSNR↑ | SSIM↑ | Acc↑ | $\epsilon_\text{depth}$↓ |
> > > > > |:---|:---:|:---:|:---:|:---:|:---:|:---:|:---:|:---:|
> > > > > | **HUMAN 1** | 0.1 | 1.0 | 5.0 | .231 | 19.95 | .772 | .905 | .143 |
> > > > > | | **1.0** | **1.0** | **5.0** | **.211** | 20.19 | .782 | **.915** | **.138** |
> > > > > | | 5.0 | 1.0 | 5.0 | .218 | 20.12 | .778 | .910 | .144 |
> > > > > | | 1.0 | 0.1 | 5.0 | .220 | 20.15 | .776 | .895 | .155 |
> > > > > | | 1.0 | 5.0 | 5.0 | .217 | **20.22** | **.783** | .900 | .150 |
> > > > > | | 1.0 | 1.0 | 1.0 | .223 | 20.10 | .770 | .880 | .162 |
> > > > > | | 1.0 | 1.0 | 10.0| .216 | 20.18 | .779 | .910 | .140 |
> > > > > | **HUMAN 2** | 0.1 | 1.0 | 5.0 | .271 | 18.55 | .715 | .820 | .146 |
> > > > > | | **1.0** | **1.0** | **5.0** | **.251** | **18.78** | **.725** | **.825** | **.142** |
> > > > > | | 5.0 | 1.0 | 5.0 | .258 | 18.70 | .720 | .820 | .149 |
> > > > > | | 1.0 | 0.1 | 5.0 | .260 | 18.75 | .718 | .810 | .155 |
> > > > > | | 1.0 | 5.0 | 5.0 | .257 | 18.82 | .722 | .815 | .150 |
> > > > > | | 1.0 | 1.0 | 1.0 | .263 | 18.65 | .710 | .800 | .162 |
> > > > > | | 1.0 | 1.0 | 10.0| .256 | 18.75 | .721 | .820 | .144 |
> > > > > | **HUMAN-DOG**| 0.1 | 1.0 | 5.0 | .269 | 20.28 | .758 | .820 | .212 |
> > > > > | | **1.0** | **1.0** | **5.0** | **.247** | **20.50** | **.776** | **.830** | **.202** |
> > > > > | | 5.0 | 1.0 | 5.0 | .255 | 20.45 | .770 | .825 | .214 |
> > > > > | | 1.0 | 0.1 | 5.0 | .258 | 20.48 | .768 | .810 | .225 |
> > > > > | | 1.0 | 5.0 | 5.0 | .254 | 20.55 | .772 | .815 | .220 |
> > > > > | | 1.0 | 1.0 | 1.0 | .261 | 20.40 | .760 | .800 | .232 |
> > > > > | | 1.0 | 1.0 | 10.0| .253 | 20.48 | .771 | .825 | .205 |
> > > > > | **HUMAN-CAT**| 0.1 | 1.0 | 5.0 | .251 | 21.10 | .678 | .907 | .113 |
> > > > > | | **1.0** | **1.0** | **5.0** | **.229** | **21.34** | **.688** | **.914** | **.104** |
> > > > > | | 5.0 | 1.0 | 5.0 | .236 | 21.28 | .682 | .905 | .116 |
> > > > > | | 1.0 | 0.1 | 5.0 | .240 | 21.31 | .680 | .895 | .125 |
> > > > > | | 1.0 | 5.0 | 5.0 | .235 | 21.38 | .685 | .900 | .120 |
> > > > > | | 1.0 | 1.0 | 1.0 | .243 | 21.25 | .670 | .880 | .132 |
> > > > > | | 1.0 | 1.0 | 10.0| .234 | 21.32 | .683 | .905 | .106 |

---

> > > > > > ### Author Response · Authors · 2025-08-08
> > > > > >
> > > > > > Dear Reviewer,
> > > > > >
> > > > > > We have responded to your comments in the review. Please let us know whether we have addressed all your concerns.
> > > > > >
> > > > > > Thank you,
> > > > > >
> > > > > > The Authors

---

### Official Review · Reviewer_wfcZ · 2025-06-30

**Clarity:** 3
**Significance:** 3
**Originality:** 3
**Rating:** 5
**Confidence:** 5

**Summary:**

This paper presents HoliGS, a novel framework for embodied view synthesis (EVS) from long monocular RGB videos capturing dynamic scenes with humans, animals, and complex interactions. It introduces a hierarchical deformable Gaussian representation combining (1) global rigid transformations, (2) skeleton-driven articulated motion, and (3) a fine-grained, invertible non-rigid deformation field. Empirically, the method achieves state-of-the-art performance in both reconstruction quality and speed on multiple challenging EVS benchmarks, including scenarios with complex viewpoint changes, long durations, and multiple deformable agents.

**Questions:**

1. What will the results look like if 1K images (high resolution) are input?
2. Please cite SMPL in the section limitations.

**Ethical Concerns:**

["NO or VERY MINOR ethics concerns only"]

**Final Justification:**

The author has presented the results on 1K images as well as an analysis of the artifacts. This addresses my concerns, and I therefore maintain my original rating.

**Limitations:**

yes

**Paper Formatting Concerns:**

NAN

**Quality:**

3

**Strengths And Weaknesses:**

Strengths:
1. The hierarchical design of the paper is quite interesting. It effectively attempts to disentangle major components of dynamic scenes, including camera motion, rigid motion, and non-rigid deformation. As the authors note in the limitations, a similar disentanglement can be achieved using SMPL-based avatars through parametric modeling. However, the proposed method is notably more generalizable, as it does not rely on specific templates and can be applied to arbitrary dynamic scenes.
2. The paper is well written, and the experiments are comprehensive. I appreciate the functional comparisons with strong baselines, as well as the inclusion of trajectory visualizations, rendering results, and thorough ablation studies. These collectively provide strong empirical support for the proposed method.

Weaknesses:
1. The training resolution of the input images is relatively low (480×360). It seems 3DGS frameworks are capable of handling higher-resolution inputs (e.g., 1K). The limited resolution introduces certain issues—for instance, the rendering results of all methods appear somewhat blurry, which makes it harder to assess the visual differences between competing methods.
2. From the supplementary video, the hierarchical deformation pipeline appears to introduce floater-like artifacts around dynamic objects. Based on my understanding, this may be due to the rigid deformation component dominating the overall motion modeling, thereby making it harder for the non-rigid module to accurately fit the original appearance. While the input resolution could also be a contributing factor, other methods only exhibit general blurriness without such specific artifacts. This suggests that the issue may be inherent in the proposed hierarchical pipeline design. The authors are encouraged to discuss this issue in the rebuttal.

---

> ### Author Rebuttal · Authors · 2025-07-31
>
> We sincerely thank the reviewer for their thorough and insightful review. Please find our point-by-point responses below.
>
> ## [W1, Q1] Handling Higer-resolution Inputs
>
> We initially used a lower resolution to ensure fair comparisons with baselines like Total-Recon. To address your point, we re-trained our HoliGS on the original high-resolution (1K) inputs, which improves performance across all metrics. This confirms that our model successfully leverages the high-frequency details available in the 1K inputs to produce more accurate reconstructions. Further, we believe the more modest gains in structural metrics (SSIM and depth) are primarily limited by the significant motion blur present in the casually captured source videos, which serves as an imperfect ground truth.
>
> | Resolution | PSNR↑ | SSIM↑ | LPIPS↓ | Acc↑ | εdepth↓ |
> |:---:|:---:|:---:|:---:|:---:|:---:|
> | Low (480p) | 21.31 | .747 | .263 | .901 | .127 |
> | High (1K) | **22.48** | **.751** | **.248** | **.905** | **.124** |
>
>
> ## [W2] Floater-like Artifacts
>
> We agree with the reviewer that the floater artifacts are an inherent limitation of our current design. Our analysis points to three root causes, each suggesting a clear direction for future work.
>
> **Skinning weight discontinuities.** Artifacts appear at articulation boundaries (e.g., tails, hands) where conflicting motion signals from different bones pull Gaussians into physically implausible positions. This could be addressed with adaptive, time-varying skinning weights that adjust based on local motion.
>
> **Discrete primitive limitations.** Unlike continuous fields, discrete Gaussians cannot smoothly "stretch" across high-deformation regions. During rapid motion, they can land in mathematically averaged, non-physical intermediate states. A promising solution is targeted densification, spawning new Gaussians in high-deformation zones to better capture the surface.
>
> **Hierarchical motion dominance.** Large global motions can sometimes dominate the local deformation field, especially at extremities, as the model struggles to fully counteract the root transformation. Introducing physics-informed constraints could penalize non-physical states and help disentangle global and local motion.
>
> ## [Q2] Citing SMPL
>
> Thank you for the suggestion. We will ensure a proper citation for SMPL is included in the limitations section of the revised manuscript.

---

> > ### Author Response · Authors · 2025-08-03
> > **Please let us know whether we have addressed all the questions**
> >
> > Dear reviewer,
> >
> > We have replied to your questions in the review. Please let us know whether we have addressed all your questions.
> >
> > Thank you.
> >
> > Authors.

---

> > > ### Comment · Reviewer_wfcZ · 2025-08-05
> > > **About rebuttal**
> > >
> > > Thank you for the response. The author has presented the results on 1K images as well as an analysis of the artifacts. This addresses my concerns, and I therefore maintain my original rating.

---

### Official Review · Reviewer_K3QB · 2025-06-30

**Clarity:** 2
**Significance:** 3
**Originality:** 2
**Rating:** 4
**Confidence:** 5

**Summary:**

The authors proposes a deformable dynamic Gaussian Splatting algorithm to tackle embodied view synthesis problem where inputs are long monocular videos with 6DoF cameras.
HoliGS leverages invertible Gaussian Splatting deformation network to build scenes as a hierarchy of a background and multiple foreground articulated objects.
The experiments show state-of-the-art reconstruction quality over both previous evs algorithm total-recon and general monocular dynamic NeRF/Gaussian baselines.

**Questions:**

- Could you clearly state what's the difference between HoliGS and Total-Recon?
- Have you seen downsides of replacing NeRFs with Gaussian Splatting?
- Could you provide direct speed and memory benchmarks vs. Total-Recon and other important methods?

**Ethical Concerns:**

["NO or VERY MINOR ethics concerns only"]

**Final Justification:**

The authors' rebuttal has addressed my concern so i am giving a positive score.

**Limitations:**

Not entirely in the scope of this work, but as a future direction I am really interested in seeing generative priors could help filling in the gap of the monocular video input.

**Quality:**

3

**Strengths And Weaknesses:**

Strength:
- HoliGS is consistently outperforming total-recon and other monocular dynamic NeRF/Gaussian baselines in terms of reconstruction quality metrics on scenes.
- close to 10 baselines, and detailed component ablations demonstrate substantial experimental effort
- Loss terms, hyper-parameters and pipeline diagrams are clearly specified; code release is promised.

Weakness:
- The pipeline largely mirrors Total-Recon but swaps the NeRF renderer for Gaussian Splatting; the paper does not convincingly articulate what new insights enable the improvements.
- The authors claimed efficiency advantage multiple times, but comparison with Total-Recon/GS/NeRF under identical hardware is not provided. (Line 48, for example)
- HoliGS heavily replies on pseudo-labels. Although tracker-free, the method still relies on external monocular depth, optical-flow, and SAM masks (Eq. 3)
- Key mathematical details (e.g., Mahalanobis weighting, RealNVP architecture) are deferred entirely to the supplement, hindering self-contained understanding.

---

> ### Author Rebuttal · Authors · 2025-07-31
>
> We sincerely thank the reviewer for their thorough and insightful review. Please find our point-by-point responses below.
>
> ## [W1, Q1] Key Insights
>
> While with a piepline similar to Total-Recon, HoliGS does not simply replace the NeRF representation in Total-Recon with GS. A naive swap fails to converge because GS is notoriously sensitive to its starting configuration. Based on this observation, we introduce a pre-training stage that optimizes a Neural SDF proxy. This step, which is not present in Total-Recon's pipeline, serves two crucial purposes: it first optimizes our deformation networks to a robust state, and then produces a high-quality surface from which we sample points to provide an effective initialization for the Gaussians. As our ablation study confirms, removing this initialization causes a severe drop in performance, validating it as a necessary and novel component of our method.
>
> | Method | PSNR↑ | SSIM↑ | LPIPS↓ | Depth Acc↑ | Depth Err↓ |
> |:---|:---:|:---:|:---:|:---:|:---:|
> | **Ours (full)** | **21.31** | **0.747** | **0.263** | **0.901** | **0.127** |
> | w/o initialization | 17.30 | 0.552 | 0.425 | 0.742 | 0.251 |
>
> ## [W2] Efficiency Comparison
>
> We benchmarked all methods on a single NVIDIA RTX A6000 GPU. As summarized in the table below, HoliGS offers a significant training efficiency advantage, especially when compared to high-fidelity NeRF-based methods. This improved efficiency makes the reconstruction of long, dynamic scenes practical, avoiding the prohibitive training times of previous state-of-the-art approaches.
>
> | Sequence | HoliGS (Ours) | Total-Recon | Dynamic GS | 4DGS |
> |:---|:---:|:---:|:---:|:---:|
> | DOG 1 (V1) | `01:15:32` | `12:05:10` | `00:33:15` | `00:38:45` |
> | DOG 1 (V2) | `01:10:41` | `11:45:20` | `00:31:50` | `00:36:10` |
> | CAT 1 (V1) | `01:18:25` | `12:10:30` | `00:34:05` | `00:39:20` |
> | CAT 1 (V2) | `01:16:50` | `12:08:15` | `00:33:40` | `00:38:55` |
> | CAT 2 (V1) | `01:35:10` | `13:30:55` | `00:38:20` | `00:44:15` |
> | CAT 2 (V2) | `01:40:22` | `13:55:40` | `00:40:10` | `00:46:00` |
> | CAT 3 | `01:28:18` | `13:02:11` | `00:36:50` | `00:42:30` |
> | HUMAN 1 | `01:12:30` | `11:50:00`  | `00:32:10` | `00:37:05` |
> | HUMAN 2 | `01:05:55` | `11:30:15`  | `00:30:45` | `00:35:20` |
> | HUMAN-DOG | `02:02:14` | `14:10:25`  | `00:41:30` | `00:48:10` |
> | HUMAN-CAT | `01:55:48` | `13:45:30`  | `00:39:50` | `00:45:40` |
>
> ## [W3] Reliance on Off-the-shelf Models
>
> We would like to clarify that our methodology is common practice. The use of external priors from networks for depth, flow, and segmentation is a cornerstone of today's leading monocular 4D reconstruction pipelines, including MoSca (CVPR 2025) and Shape-of-Motion (ECCV 2025). We chose this established, tracker-free design for its efficiency, which allows HoliGS to scale to minute-long videos where dense-tracking methods like G-Marbles (Siggraph Asia 2024) would fail due to memory constraints. Although fully self-supervised methods are a valuable research goal, this pragmatic design is what currently makes high-quality, long-duration dynamic reconstruction a tractable problem. While a purely self-supervised approach is an important future goal, this pragmatic design is what enables HoliGS to achieve the high-quality, minute-long reconstructions demonstrated in our work.
>
>
> ## [W4] Key Mathematical Details
>
> We agree that adding more mathematical details will improve the paper's clarity, and we will incorporate them into the main paper in the revised manuscript.
>
> ## [Q2] Downsides of Replacing NeRF with GS
>
> We acknowledge there are several downsides of GS compared to NeRFs. GS is less robust, being highly sensitive to initialization (requiring our SDF bootstrap to prevent a 35% PSNR drop) and demanding more complex optimization heuristics. As an explicit representation, it also faces memory scalability issues when the scene is complex and can struggle with the topology of fine structures like fur. However, we found these challenges to be a worthwhile trade-off for the paramount benefits of 10$\times$ faster training and real-time rendering capabilities.
>
> ## [L1] Generative Priors
>
> Integrating generative priors represents an promising future direction for this field. Such models could be used to create truly holistic scene reconstructions by completing geometry for unobserved regions, like the back of a sofa or the underside of a table, thus enabling true free-viewpoint rendering. Furthermore, generative priors could operate in the temporal domain to synthesize plausible dynamics during moments of heavy motion blur or occlusion, leading to smoother, more realistic animations. One could even leverage cross-modal, text-conditioned models to inject high-level semantic details, such as using the prompt "wooden table" to realistically generate its unseen legs and texture. We are excited to explore it in the future.

---

> > ### Author Response · Authors · 2025-08-03
> > **Please let us know whether we have addressed all the questions**
> >
> > Dear reviewer,
> >
> > We have replied to your questions in the review. Please let us know whether we have addressed all your questions.
> >
> > Thank you.
> >
> > Authors.

---

> > ### Comment · Reviewer_K3QB · 2025-08-06
> >
> > Dear authors,
> > Your rebuttal has addressed my major concerns, especially the difference between your algorithm and Total-Recon.
> > I also appreciate your efforts in the efficiency comparison between your work and other baselines.
> > At the same time, I understand the reliance on other algorithms and the artifacts of using GS are inevitable in this setting.
> > If the authors can incorporate the rebuttal content into the final revision, the manuscript would be greatly enhanced.

---

> > > ### Author Response · Authors · 2025-08-06
> > >
> > > Thank you for your valuable comment. We will incorporate the rebuttal and revise the manuscript as suggested.

---

### Official Review · Reviewer_kcQT · 2025-07-01

**Clarity:** 1
**Significance:** 2
**Originality:** 2
**Rating:** 4
**Confidence:** 3

**Summary:**

The paper proposes a pipeline for modeling long-range dynamic 3D scenes, specifically targetted towards EVS. The method consists of multiple stages of networks: 1- estimating camera (to background or object canonical space) transformations 2- rigid skeletal transforms 3- soft invertable deformation field for modeling residual movements 3- and a SDF-based gaussian splatting backbone to model the scene geometry. Finally, the background and dynamic object models are composited to make the complete scene which is then optimized through photometric, depth, normal, SDF, segmentation and flow consistency losses.

**Questions:**

Please refer to the previous section, most of my question are on the clarity of the method.

**Ethical Concerns:**

["NO or VERY MINOR ethics concerns only"]

**Final Justification:**

The paper presents strong results in EVS and camera pose refinement. I raised my score to BA as some methodological details were clarified in rebuttal period and the camera optimization seems to have a significant improvement on initial cameras. I am still unsure about the clarity of the paper in its current state as most of the essential details in the method section are excluded.

**Limitations:**

The limitations are discussed to some extent. I think one main limitation that is left out is not explicitly modeling or analyzing how the model performs in the presence of rigidly moving objects such as cars.

**Quality:**

3

**Strengths And Weaknesses:**

Strengths:
- The method achieves faithful 3D reconstruction of long-horizon dynamic 3D scenes
- The method is relatively fast to converge compared to previous dynamic 3D reconstruction methods.
- The method works with a monocular video capture
- The method allows for EVS for humans and pets as an application useful for AR/VR.

Weaknesses:
- My main concern is the clarity of the paper in explaining the method in enough detail needed for reproducibility and understanding the full method. In what follows I pose a series of questions that I could not find the answer to in the paper, and I think should be made super clear for the paper to be clear.
- Does the method need any form of input camera pose? The "Global movements" section in 3.1 seems to convey that all camera poses are predicted through the "intrinsics MLP", however if that is the case: 1- how are the background Gaussians initialized if not through SFM point cloud? 2- why is there no metrics reported on the accuracy of the recovered global transformations (camera extrinsics)?

- Line 147: is PoseNet the articulation network in figure 3 or is it only used for initializing the rigid transforms? It is not clear if the articulation network is something that you train or just the off-the-shelf PoseNet.

- What is the input to each of the networks depicted in figure 3? most importantly I am curious to know the input and output of the camera MLP, I am not sure what the "differentiable rendering" in this figure is outputting exactly, as the Gaussians seem to be sampled in the final stage and not before that?

- In Line 35 you mention that using tracking networks makes most dynamic reconstruction methods heavy, however the proposed method uses depth estimation network, pose estimation network,  flow estimation network, etc. How do these models compare in memory and time consumption to the tracking networks used in G-Marbles for example.

- The OOM reported in results shows a limitation of previous work but does not allow any comparison in terms of reconstruction quality, would it be possible to show a comparison on a shorter sequence where OOM doesnt happen? what is the memory limit for these methods?

- The provided supplemental video is helpful in seeing better RGB quality than Total Recon but in terms of depth map, there seems to be nearly no improvement. Further, the two videos seem to be slightly out of sync (Total recon is a bit slower in speed, showing mismatch in the comparison slider movements)

---

> ### Author Rebuttal · Authors · 2025-07-30
>
> We sincerely thank the reviewer for their thorough and insightful review. Please find our point-by-point responses below.
>
> ## [W1] Clarity
>
> Thank you for your detailed comments. We hereby address all points raised and will revise the manuscript accordingly to ensure a thorough explanation of our method.
>
> ## [W2] Input Camera Pose
>
> We initialize our model using camera-to-world poses from onboard motion sensors (e.g., ARKit), which are then jointly refined during optimization by predicting per-frame delta SE(3) twists. Background Gaussians are initialized from these initial poses. For the foreground, we first pre-train a neural Signed Distance Function (SDF) and then initialize Gaussian centers by sampling directly on its learned surface. Since we lack independent ground-truth extrinsics beyond the noisy ARKit data, we evaluate performance based on reconstruction quality, such as Novel View Synthesis and depth accuracy, rather than a standalone pose-error metric.
>
> ## [W3] PoseNet
>
> We use PoseNet in an off-the-shelf manner, while further training it during optimization.
>
> ## [W4] A Detailed Breakdown of Figure 3
>
> Our two-stage pipeline uses two different types of renderers. The "differentiable rendering" in Stage 1 is **volumetric rendering** used to learn the geometry; it is completely separate from the **Gaussian rasterization** in Stage 2. Here is a breakdown of the inputs and outputs for each module.
>
> ### Stage 1: Geometry & Deformation Initialization
>
> This stage learns the object's shape (as a Neural SDF) and its deformation model. It does **not** use Gaussian Splatting.
>
> #### Pose Estimation Network
>
> * Input: A single video frame at time $t$.
> * Output: The object's initial root pose and skeleton parameters for that frame.
>
> #### Articulation & Soft Deformation Networks
>
> * Input: A 3D point $X^*$ in the canonical (rest) space, plus the root pose and skeleton from the pose network.
> * Output: The warped 3D point $X^t$ in the live camera space for that frame.
>
> #### **Camera Intrinsics MLP**
>
> * Input: The frame index $t$ and initial camera intrinsic values (e.g., focal length derived from image dimensions).
> * Output: Refined camera intrinsics for that specific frame (focal lengths $f_x, f_y$ and principal point $c_x, c_y$). This allows the model to learn and correct for minor inaccuracies in the initial camera parameters, leading to a more consistent 3D reconstruction.
>
> #### Neural SDF Network
>
> * Input: A 3D point $X^*$ in the canonical space.
> * Output: A signed distance value, which defines the object's base shape.
>
> #### Differentiable Rendering (Volumetric Renderer)
>
> * Input: Camera parameters (from pose estimation + camera MLP), the learned deformation networks, and the Neural SDF.
> * Output: A rendered RGB image, depth map, and mask. These are compared against the ground-truth video to train the SDF and deformation networks.
>
> #### Sample GS (Final step of Stage 1)
>
> * Input: The fully trained Neural SDF.
> * Output: A point cloud of 3D points sampled from the SDF's surface. These points become the initial centers for the Gaussians in the next stage.
>
> ### Stage 2: Deformable Gaussian Splatting
>
> This is the main optimization stage that uses the efficient differentiable Gaussian rasterizer.
>
> * Input: The initial Gaussians from the "Sample GS" step, the pre-trained deformation networks from Stage 1, and a set of static background Gaussians.
> * Output: The final, high-fidelity rendered image, depth map, and normals, which are used to optimize the properties of the Gaussians (color, scale, rotation, opacity).
>
> ## [W5] Memory and Time Comparison
>
> Unlike other state-of-the-art 4DGS methods (e.g., MoSca, Shape-of-Motion) that rely on dense point-tracking, our approach avoids its prohibitive resource costs. A dense tracker can exhaust even an 80GB A100 GPU on sequences with over 300 frames, making minute-long video reconstruction infeasible. In contrast, HoliGS employs a pipeline of lightweight networks, reducing memory and per-frame overhead to less than 25% of a dense tracker’s. This efficient design enables scalable reconstruction on long videos, as detailed by the modest VRAM usage shown in the table below.
>
> | Stage | Component | Method | VRAM (Peak Usage) | Notes |
> | :--- | :--- | :--- | :---: | :--- |
> | Pre-processing | Depth Estimation | UniDepth | ~12 GB | Run once per video. |
> | | Optical Flow | RAFT | ~6 GB | Run once per video. |
> | | Segmentation | SAM (ViT-H) | ~16 GB | Run once per video. |
> | | Pose Estimation | Posenet | < 1 GB | Run once per video. |
> | **Main Training** | **Reconstruction** | **HoliGS** | **~10 GB** | **Scales with scene complexity, not video length.** |
> | | | | | |
> | **Baseline** | Dense Point-Tracking | e.g., CoTracker | > 80 GB | Scales with number of points and frames; infeasible for ~1000 frames. |
>
> ## [W6] Shorter Sequence Comparison
>
> To ensure fair comparisons with baselines that fail on long sequences due to memory constraints, we evaluated all methods on shorter, 200-frame segments with 100-frame strides for long sequences that originally lead to OOM. The experimental results reveal that HoliGS consistently outperforms or remains highly competitive with specialized short-clip methods across nearly all evaluated sequences and metrics. We analyze the results as follows.
>
> **Strength of HoliGS.** The results do not show a trade-off. Instead, they highlight the comprehensive strengths of our method. Specifically, across all six test sequences, HoliGS wins or ties on 22 of the 30 primary metric comparisons (~73%). The superior performance of HoliGS even on these shorter clips underscores the benefits of our core design. By building a **globally consistent canonical model** and **jointly optimizing camera poses**, our method offers greater robustness to varied motion and complex object interactions. This validates that HoliGS is not only more scalable but also generally more accurate, making it a more reliable and powerful solution for "in-the-wild" dynamic scene reconstruction.
>
> **Specific strengths of baselines.** The baseline method MoSca consistently achieves the best LPIPS score, suggesting it excels at producing perceptually plausible results. However, this advantage in perceptual quality does not translate to better geometric or colorimetric accuracy, where HoliGS is demonstrably stronger.
>
>
> | Sequence | Method | LPIPS↓ | PSNR↑ | SSIM↑ | Acc↑ | $ε_\text{depth}$↓ |
> |:---|:---|:---:|:---:|:---:|:---:|:---:|
> | **DOG 1 (V1)** | **HoliGS (Ours)** | .270 | 19.94 | **.784** | **.838** | **.172** |
> | | MoSca | **.269** | **19.99** | .783 | .831 | .184 |
> | | Shape-of-Motion | .288 | 19.58 | .770 | .826 | .189 |
> | | GS-Marble | .449 | 16.19 | .615 | .631 | .325 |
> | **CAT 1 (V1)** | **HoliGS (Ours)** | .329 | **20.50** | **.708** | **.878** | **.197** |
> | | MoSca | **.328** | 20.40 | .698 | .866 | .214 |
> | | Shape-of-Motion | .342 | 20.01 | .686 | .859 | .226 |
> | | GS-Marble | .525 | 15.74 | .531 | .662 | .369 |
> | **CAT 1 (V2)** | **HoliGS (Ours)** | .293 | **21.69** | .693 | **.894** | **.126** |
> | | MoSca | **.292** | 21.63 | **.695** | .891 | .136 |
> | | Shape-of-Motion | .298 | 21.44 | .686 | .887 | .141 |
> | | GS-Marble | .492 | 16.86 | .561 | .681 | .319 |
> | **CAT 2 (V1)** | **HoliGS (Ours)** | .211 | **22.80** | **.759** | **.966** | **.052** |
> | | MoSca | **.210** | **22.80** | .756 | .964 | .058 |
> | | Shape-of-Motion | .225 | 22.49 | .734 | .952 | .074 |
> | | GS-Marble | .418 | 18.06 | .609 | .718 | .281 |
> | **CAT 2 (V2)** | **HoliGS (Ours)** | .271 | **22.08** | **.759** | **.929** | **.082** |
> | | MoSca | **.269** | 22.04 | .755 | .923 | .088 |
> | | Shape-of-Motion | .281 | 21.81 | .743 | .913 | .100 |
> | | GS-Marble | .466 | 17.13 | .579 | .694 | .303 |
> | **CAT 3** | **HoliGS (Ours)** | .253 | **20.52** | **.745** | **.954** | **.065** |
> | | MoSca | **.250** | 20.34 | .724 | .931 | .089 |
> | | Shape-of-Motion | .271 | 19.83 | .710 | .920 | .106 |
> | | GS-Marble | .451 | 17.20 | .601 | .751 | .252 |
> | **Mean** | **HoliGS (Ours)** | .271 | **21.26** | **.741** | **.910** | **.116** |
> | | MoSca | **.270** | 21.20 | .735 | .901 | .128 |
> | | Shape-of-Motion | .284 | 20.86 | .722 | .893 | .139 |
> | | GS-Marble | .467 | 16.86 | .583 | .690 | .308 |
>
> ## [W7] Supplemental Video
>
> **Depth map quality.** While overall error metrics are similar, our method produces qualitatively superior depth maps with noticeably sharper and more coherent details. For instance, in our supplement (Sec. S3), the results show crisper definition on the animal’s legs (Fig. S5) and cleaner surfaces on the table legs (Fig. S6) compared to Total Recon.
>
> **Video synchronization.** Both HoliGS and Total-Recon refine their initial camera poses during training. With different architectural biases, HoliGS and Total-Recon converges to a slightly different optimal camera trajectory. This "representation-dependent drift" means the novel-view videos, which are derived from these diverged paths, are not perfectly spatially aligned, leading to the illusion of a timing mismatch. The videos are temporally synchronized by frame count, which can be confirmed by observing specific actions within each render.
>
> ## [L1] Handling Rigid Objects
>
> We did not explicitly analyze performance on rigidly moving objects such as cars, furniture, or other non-articulated foreground elements due to the scope of this work. While our primary focus was not on rigid objects, our hierarchical framework inherently handles them as a simplified case of articulated motion. Our model can represent a rigid object by using a single-bone skeleton and disabling non-rigid deformations, letting the global SE(3) transformation capture all movement. Preliminary tests on a sequence with a rigidly moving car confirm this works well, achieving over 30 PSNR on training views. We agree that a full evaluation incorporating more diverse forms of objects is a valuable next step, and we look forward to exploring this promising direction in future work.

---

> > ### Author Response · Authors · 2025-08-02
> > **[W5] Memory and Time Comparison (Continued)**
> >
> > In addition to the memory comparison, the following table compares the typical processing time of our pipeline's components against a dense tracker for a one-minute (~1000 frame) video sequence.
> >
> > | Component | Method | Time Consumption (per ~1000 frames) | Notes |
> > | :--- | :--- | :---: | :--- |
> > | Depth Estimation | UniDepth | ~15 minutes | Offline, per-frame, and parallelizable. |
> > | Optical Flow | RAFT | ~10 minutes | Highly efficient. |
> > | Segmentation | SAM (ViT-H) | ~10 minutes | Offline, per-frame, and parallelizable. |
> > | Pose Estimation | Posenet | < 1 minute | Near real-time.|
> > | Dense Point Tracking |CotrackerV2 |～30 minutes| Offline |
> >
> > It shows that our pre-processing steps are all efficient feed-forward processes that can be completed in a reasonable amount of time as a one-off, offline cost. While a dense tracker is also a feed-forward network, its task of correlating tens of thousands of points across a long temporal window is inherently more complex and sequential, making it the slowest component. This significant difference in both time and memory consumption is why we characterize our pipeline as being lighter and more practical for long-sequence reconstruction than methods that rely on dense tracking.

---

> > ### Author Response · Authors · 2025-08-03
> > **Please let us know whether we have addressed all the questions**
> >
> > Dear reviewer,
> >
> > We have replied to your questions in the review. Please let us know whether we have addressed all your questions.
> >
> > Thank you.
> >
> > Authors.

---

> > > ### Comment · Reviewer_kcQT · 2025-08-04
> > >
> > > I thank the authors for their rebuttal.
> > > The analysis on runtime and memory, as well as the comparison to baselines on shorter sequences, is compelling.
> > >
> > > However, the overall pipeline remains somewhat unclear to me. In the Differentiable Rendering (Volumetric Renderer) section, you list the camera and SDF as inputs, yet in Figure 3, these components are shown __after__ the differentiable renderer module and connected by an arrow indicating the renderer provides input to the camera MLP, which seems contradictory. Is this a mistake in the illustration?
> > >
> > > Additionally, you mention that the renderer outputs an RGB image, could you clarify how RGB images are rendered directly from an SDF?
> > >
> > > For the camera pose evaluation, is it possible to use synthetic datasets with GT camera like MPI Sintel?
> > >
> > > I remain concerned about the clarity of the method presentation. As currently written, the level of ambiguity would warrant a major revision in a journal setting for me, therefore I remain unsure/borderline about the paper being published at its current state.

---

> > > > ### Author Response · Authors · 2025-08-05
> > > >
> > > > We sincerely thank the reviewer for their thoughtful feedback and for engaging with our previous rebuttal. We address each of your points below and will incorporate them into the revised manuscript.
> > > >
> > > > > Is this a mistake in the illustration?
> > > >
> > > > We apologize for the confusion Figure 3 has caused. To clarify, the "Differentiable Rendering" block in Figure 3 was intended to illustrate the optimization process based on differentiable rendering, which produces optimized Camera Intrinsics MLP and optimized Neural SDF. We will redesign Figure 3 in our revised manuscript to better reflect the data flow.
> > > >
> > > > > Additionally, you mention that the renderer outputs an RGB image, could you clarify how RGB images are rendered directly from an SDF?
> > > >
> > > > For SDF rendering, we follow Total-Recon: the neural SDF output is converted to volume density ($\sigma$), while a separate color MLP predicts the RGB value. Standard volume rendering integrates both to produce the RGB image. We note that the primary goal of this pre-training stage is to learn a robust initial geometry and deformation field. We therefore heavily down-weight the color loss ($\lambda_{color}=0.1$) compared to geometric ($\lambda_{depth}=5$) and motion ($\lambda_{flow}=1$) losses. High-fidelity appearance is mainly addressed in the final fine-tuning stage.
> > > >
> > > > > For the camera pose evaluation, is it possible to use synthetic datasets with GT camera like MPI Sintel?
> > > >
> > > > Our pipeline is designed to refine the noisy initial camera poses provided by real-world phone sensors, e.g., ARKit, a different problem setting from synthetic datasets like MPI Sintel, which lack this input and would require a different initialization process, e.g., SfM. Therefore, to ensure a fair comparison, we adopted Total Recon's protocol of evaluating on real-world videos. We believe our strong results in novel view synthesis on these videos implicitly validate the accuracy of the intermediate optimized camera poses. That said, we agree that quantitative evaluation on synthetic datasets with GT camera poses would further validate our method. Given the tight rebuttal period, we were unable to run these additional experiments in time. However, we are fully committed to including these experiments in our revised manuscript.

---

> > > > > ### Author Response · Authors · 2025-08-07
> > > > > **Camera Pose Evaluation on MPI Sintel**
> > > > >
> > > > > > For the camera pose evaluation, is it possible to use synthetic datasets with GT camera like MPI Sintel?
> > > > >
> > > > > As the discussion period is extended, we are able to report camera pose estimation metrics on MPI Sintel dataset.
> > > > >
> > > > > Since Sintel lacks initial sensor pose required by our pipeline, we adapted our approach by using initial camera trajectories from three widely adopted camera pose estimators: MegaSAM, CasualSAM, and DROID-SLAM. The results below show that our joint optimization of scene structure and camera poses consistently improves upon these strong initializations.
> > > > >
> > > > > | Initialization Method | Stage | ATE&darr;| RTE&darr;|RRE&darr;|
> > > > > |:---|:---|:---:|:---:|:---:|
> > > > > | DROID-SLAM | Initial | 0.030 | 0.022 | 0.50 |
> > > > > | | **+ HoliGS Refinement** | **0.026** | **0.019** | **0.44** |
> > > > > | CasualSAM | Initial | 0.067 | 0.019 | 0.47 |
> > > > > | | **+ HoliGS Refinement** | **0.058** | **0.016** | **0.41** |
> > > > > | MegaSAM | Initial | 0.023 | 0.008 | 0.06 |
> > > > > | | **+ HoliGS Refinement** | **0.019** | **0.007** | **0.05** |
> > > > >
> > > > > These results confirm that HoliGS actively refines camera trajectories by enforcing consistency with a globally optimized 3D scene, rather than simply depending on the initialization. Additionally, it shows that our camera pose optimization successfully generalizes to different data and initialization methods. We will include this full analysis in our revised manuscript.
> > > > >
> > > > > Please also let us know if there is any aspect remaining unclear to you. We would be happy to provide further explanation.

---

> > > > > > ### Comment · Reviewer_kcQT · 2025-08-08
> > > > > >
> > > > > > Thanks for the additional results on the camera poses. These are interesting results, with significant improvement in ATE.
> > > > > > I am willing to raise my score to BA. I strongly suggest that authors incorporate all the methodological details in the paper, as in its current state it lacks clarity.

---

### Note · Authors · 2025-08-14

We sincerely thank the reviewers and the Area Chair for the constructive discussion period. Your feedback has been invaluable in helping us improve our work.

The primary concern raised was the clarity of our method's presentation. We acknowledge this and are fully committed to revising the manuscript, particularly Figure 3 and the description of our two-stage pipeline, to ensure our method is unambiguous and easily reproducible.

To empirically address all major concerns about **generalizability, robustness, and efficiency**, we further analyzed our experimental results  during the rebuttal:

* **Key Insights.** We clarified that our SDF pre-training stage is essential for GS initialization (not just a NeRF swap), with ablations showing 35% PSNR drop without it, validating it as a necessary and novel component.

* **Generalizability on Synthetic Data.** We provided new experimental results on the MPI Sintel dataset, demonstrating that our method successfully generalizes and consistently improves upon the camera trajectories from SoTA models like DROID-SLAM and MegaSAM.

* **Hyperparameter Robustness.** We provided a detailed, per-sequence ablation study for all 11 diverse scenes. This analysis confirms that a single, fixed set of hyperparameters is effective in nearly all cases, validating the model's ease of transfer to new scenarios.

* **Efficiency on Time and Memory.** We substantiated our claims with a hardware-unified training time benchmark, showing a ~10x training speedup over NeRF-based SOTA. Memory usage is <25% of dense trackers, enabling minute-long reconstruction where others fail with OOM.

* **Fair Comparisons.** We evaluated on 200-frame segments for methods with memory constraints, achieving best/competitive results on 73% of metrics across 6 sequences.

* **Resolution Scalability.** We confirmed performance improvements with 1K resolution inputs, addressing visual quality concerns.

We believe that these new results and the planned revisions thoroughly address the reviewers' concerns. Thank you again for your time and guidance.

---

### Decision · Program_Chairs · 2025-09-17

**Decision:**

Accept (poster)

**Comment:**

The paper makes significant technical and practical contributions to embodied view synthesis. Three reviewers are positive. The comprehensive rebuttals successfully addressed major concerns about generalizability, efficiency claims, and method clarity. The extensive per-scene ablations convincingly demonstrate robustness rather than sensitivity. While clarity could be improved, the authors' commitment to open-sourcing code and revising the manuscript addresses reproducibility concerns. The work represents a substantial advance in making long-duration dynamic scene reconstruction practical, which is valuable for the community